



# OLCI A/B tandem phase: Evaluation of FLEX like radiances and estimation of systematic differences between OLCI-A and OLCI-FLEX

Lena Jänicke[1], Rene Preusker[1], Marco Celesti[2], Marin Tudoroiu[3], Jürgen Fischer[1], Dirk Schüttemeyer[4], and Matthias Drusch[4]

[1]Institute of Meteorology, Freie Universität Berlin (FUB), Carl-Heinrich-Becker-Weg 6-10, 12165 Berlin, Germany
[2]HE Space for ESA - European Space Agency, ESTEC, PO Box 299, NL-2200 AG Noordwijk, The Netherlands
[3]ESA-ESRIN, Largo Galileo Galilei 1, 00044 Frascati (Rome), Italy
[4]ESA-ESTEC, Keplerlaan 1, 2201 AZ Noordwijk, The Netherlands

**Correspondence:** Lena Jänicke (lena.jaenicke@wew.fu-berlin.de)

**Abstract.** During the tandem phase of Sentinel-3A and -3B in summer 2018 the Ocean and Land Color Imager (OLCI) mounted on Sentinel-3B satellite was reprogrammed to mimick ESA's 8th Earth explorer the Fluorescence explorer (FLEX). OLCI in FLEX configuration (OLCI-FLEX) had 45 spectral bands between 500 nm and 792 nm. The new data set with high resolution measurements (band width: 1.7-3.7 nm) serves as preparation of the FLEX mission. Co-registered measurements of

both instruments will be used to describe the atmosphere and the surface. For such combined products, it is essential that both instruments are radiometrically consistent. We developed a transfer function to compare radiance measurements from different optical sensors and to monitor their consistency.

In the presented study, the transfer function shifts information gained from high-resolution "FLEX-mode" settings to information convolved with spectral response of the normal (lower) spectral resolution of the OLCI sensor. The resulting reconstructed

low resolution radiance is representative for the high resolution data and it can be compared with the measured low resolution radiance. This difference is used to quantify systematic differences between the instruments. Applying the transfer function, we could show that OLCI-A is about 2 % brighter than OLCI-FLEX for most bands. At the longer wavelengths OLCI-A is about 5 % darker. Sensitivity studies showed that the parameters affecting the quality of the comparison of OLCI-A and OLCI-FLEX with the transfer function are mainly the surface reflectance and secondarily the aerosol composition. However, the aerosol

composition can be simplified as long it is treated consistently in all steps in transfer function.

Generally, the transfer function enables direct comparison of instruments with different spectral responses even with different observation geometries or different levels of observation. The method is sensitive to measurement biases and errors resulting from the processing. One application could be the quality control of the FLEX mission.

## 1   Introduction

Sentinel-3 is part of the European Copernicus program which provides Earth observation data for scientists and policy makers (Jutz and Milagro-Pérez, 2020). The program is designed amongst others to deliver long-term climate records. Sentinel-3





carries the Ocean and Land Color Imager (OLCI), a push-broom spectral imager with 21 bands between 400 and 1020 nm (Donlon et al., 2012). Currently, two Sentinel-3 satellites are in orbit, namely Sentinel-3A (since 2016) and Sentinel-3B (since 2018).

During the commissioning phase of Sentinel-3B in 2018, a smooth continuity was guaranteed by a tandem phase of Sentinel-3A and -3B (Clerc et al., 2020). Both satellites flew in the same orbit observing the same geographic target within 30 s. The measurements were taken with the same geometrical and environmental conditions. Thus, a comparison of the radiance data was possible (Lamquin et al., 2020).

Contributing to a deeper insight into plants activity and their response to environmental changes, ESA's eighth Earth Explorer
Fluorescence Explorer (FLEX) will be launched in 2025 (Van Wittenberghe et al., 2021). It will carry a high resolution Fluorescence Imaging Spectrometer FLORIS which measures the radiance between 500 and 780 nm (Drusch et al., 2017; Coppo et al., 2017). Its band characterization is summarized in Tab. 1. FLEX will fly in tandem formation with Sentinel 3 and the OLCI sensor (onboard of Sentinel-3) will deliver the necessary information for performing the atmospheric correction of FLORIS (Drusch et al., 2017; Coppo et al., 2017).

This tandem constellation was mimicked during the tandem phase of Sentinel-3A and -3B for 24 acquisition scenes (5 minutes each). OLCI-B was reprogrammed to measure in 45 bands between 500 and 792 nm. An overview of the spectral resolution of the different data sets is shown in Tab. 1. For a meaningful usage of the OLCI data in FLEX configuration (OLCI-FLEX), the quality of the data must be estimated. A comparison with OLCI-A is most promising as the tandem constellation allowed measurements under the same conditions. However, OLCI-FLEX has a different spectral response which does neither allow a
direct comparison nor a convolution with the spectral response of OLCI-A. To overcome this limitation, we developed a transfer function which enables the comparison of OLCI-A and OLCI-FLEX radiance measurements. It is applied for vegetated cloud free land pixels, as the main objective of FLEX mission is to retrieve fluorescence emitted by plants.

Lamquin et al. (2020) showed a systematic bias between OLCI-A and OLCI-B in the tandem constellation data. The bias of OLCI-FLEX with respect to OLCI-A will be estimated by using our transfer function. The bias should be consistent with the
findings of Lamquin et al. (2020). Furthermore, this comparison is a test of the calibration of FLORIS. Its calibration will partly rely on inter-calibration with OLCI. Niro et al. (2021) stated that Level 1 data consistency throughout the complete ESA fleet is of "utmost importance for the interoperability" of different mission products. For FLEX an inter-operational product is planned and thus a consistency with its tandem partner is necessary. This consistency is checked for OLCI-A and OLCI-FLEX using the transfer function.

An example for inter-calibration is the validation of Moderate Resolution Imaging Spectroradiometer (MODIS) Terra and Aqua shown by Angal et al. (2021). Radiative transfer simulations were used to simulate TOA reflectance based on ground-based measurements. However, for this comparison well defined surface and atmosphere descriptions are necessary. The difference in spatial resolution between ground-based and satellite-based instruments results in differences in surface description and inserts uncertainties.

The direct inter-comparison of TOA radiances is possible under the condition of simultaneous overpasses, the same spatial resolution and similar observation geometry. Under those conditions, only the spectral resolution differences must be consid-





ered. A spectral adjustment was introduced by Chander et al. (2013) who used a third high-resolution instrument to calculated a spectral band adjustment factor. However, the third instrument introduces also uncertainty, that must be determined. Furthermore, the number of samples meeting all requirements is very limited. In contrast, a comparison of instruments flying in a

tandem mission with similar spatial resolution is possible for a large number of targets allowing a robust quality control.

Our transfer function allows such comparison for instruments with the same spatial resolution but different spectral response. The application of the method on the OLCI-A/OLCI-FLEX data set enabled us to quantify a systematic bias between OLCI-A and OLCI-FLEX. The method and its application is presented in this paper. In section 2 the method is presented including the description of the input data (2.2), the radiative transfer simulations (2.4) and the 1D-variational approach (2.7). In section

3 the results are shown. We present the sources of uncertainty in section 4. In section 5 we discuss the results and draw the conclusion in section 6.

## 2 Methods

### 2.1 Description of the Transfer Function

To compare the top-of-atmosphere (TOA) radiance of OLCI-A and OLCI-FLEX, we developed a transfer function. A schematic

overview of the transfer function is given in Fig. 1. The transfer function is based on two sets of consistent radiative transfer simulations: one set simulating OLCI-FLEX and the other one OLCI-A measurements. Information about atmosphere and surface is retrieved from the higher resolution OLCI-FLEX data with an 1D variational approach (1Dvar). The information is shifted to the band characteristics of OLCI-A (light green arrows) and a forward model simulates the corresponding TOA-radiance that is based on information gained from OLCI-FLEX. The reconstructed OLCI-A radiance can be compared with

measured OLCI-A. The method is applied pixelwise. The reconstructed OLCI-A spectrum will be referred to as OLCI-AR from now on.

The used data are level 1B (L1B) data from OLCI-FLEX and OLCI-A. The L1 data include radiances, observation geometry, band characterization, inband solar irradiance, water vapour content, ozone concentration and information about the surface (sea surface pressure, altitude, temperature). In addition to the satellite data, AERONET data are used to characterize the aerosol

(see 2.2.4) (Giles et al., 2019). The input parameter for the transfer function are aerosol information, the measurement geometry and the OLCI-FLEX radiances which are gas corrected as part of the preprocessing (2.3). The core of the transfer function is the 1Dvar and the forward model which are based on look-up tables (LUTs). The output of the 1Dvar is the surface reflectance and the surface pressure. The aim of the transfer function is not to find a physically perfect state of the atmosphere and the surface but to find a state that explains as best as possible each pixel-wise measurement of OLCI-FLEX. This requirement

allows to reduce the degrees of freedom of the transfer function. The detailed description of the radiative transfer model and its input is given in sections 2.4 and 2.5.

A principle component regression (PCR) is used to shift the surface reflectance to band characteristics of OLCI-A, as described in section 2.8. The aerosol information from AERONET is shifted using linear interpolation. Together with the measurement geometry of OLCI-A, its band characteristics and the optimized surface pressure, the shifted surface and aerosol information





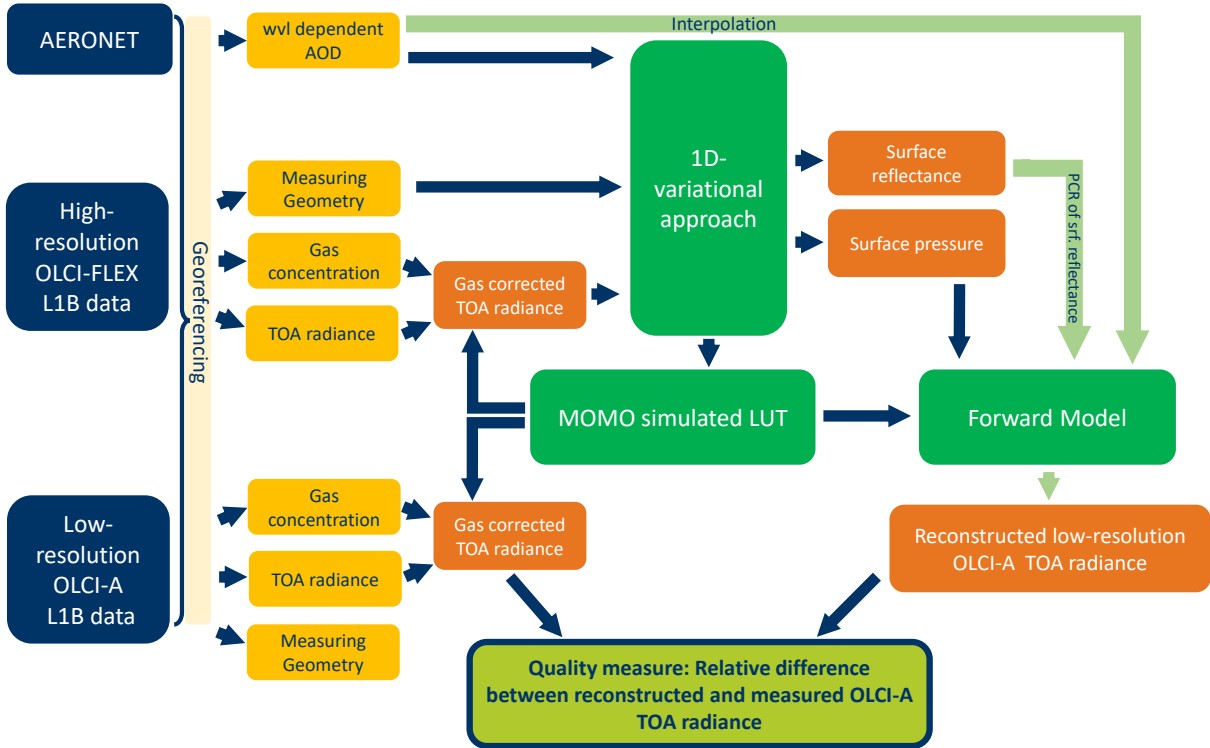

**Figure 1.** Schematic overview of transfer function for OLCI-FLEX and OLCI-A. In blue boxes: data sources, in yellow: measured input data characterizing the environment, in orange: processed data, in green processors. The light green box with blue frame is the result of the process. The dark blue arrows point in the direction of the data and the light green arrows symbolize the spectral interpolation from OLCI-FLEX bands to OLCI-A bands.

serve as input for the forward model. The resulting OLCI-AR radiance is representative for the OLCI-FLEX measurement. The difference between the reconstructed and measured OLCI-A radiance quantifies the bias between the two data sets.

## 2.2   Data

Besides the L1b radiance of OLCI-A and OLCI-FLEX additional parameters are needed as input for the transfer function. Most information are taken directly from the OLCI L1 data sets or related data like the spectral response functions for the 24

acquisition scenes. We focus on the 22 scenes over central Europe as shown in Fig. 2. In addition, two scenes were located over North America.

OLCI's spatial coverage of 1270 km is realised by 5 cameras with charged coupled devices (CCD) with 740 x 520 detectors each (Sentinel 3 CalVal Team, 2016). 740 rows are aligned across track and 520 detector rows are aligned along track. Each of the 740 detector rows has their own spectral response function per band. The detector and camera information are necessary

to identify the according spectral response function. The spectral responses of OLCI-FLEX and OLCI-A will be quantified by





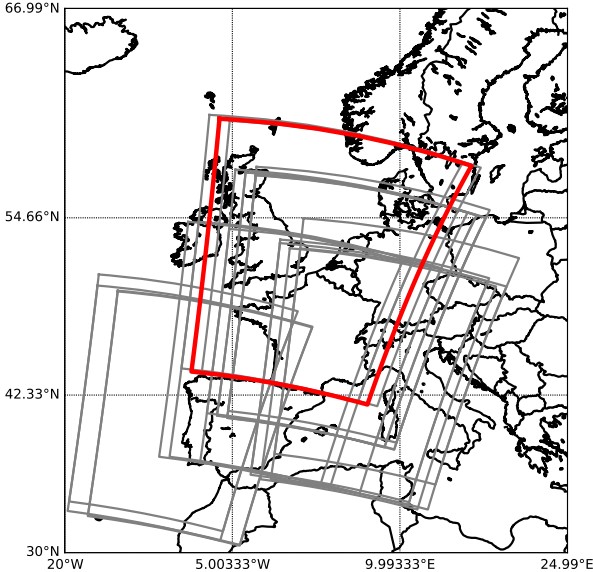

**Figure 2.** Map of Europe with frames of 22 OLCI-FLEX acquisition scenes in grey. In red, the frame of OLCI-FLEX scene on 2nd July 2018.

central wavelength and full width at half maximum (FWHM). The response functions of OLCI-A are taken from Sentinel 3 CalVal Team (2016). The ones for OLCI-FLEX were generated assuming Gaussian response of a single detector. The FWHMs of the detectors are taken from pre-launch characterization. The central wavelength are taken from the temporal evolution model of the wavelength characterization (see Sect. 2.3.3).

### 2.2.1 Pixel selection

The transfer function is applied for land pixels, because the mission is primarily designed for land applications. They are classified based on the quality flag set of OLCI-A L1-b data. Furthermore, the quality flags marking bright, invalid and saturated pixels are used from that data set. If one band is saturated in a pixel, this pixel is not used for the transfer function. Cloudy pixels are also filtered out by applying the flag bright pixel as first estimate. For a further classification of clouds the IDEPIX routine is used. It is implemented in Sentinel Application Platform SNAP (Wevers et al., 2022). Here, the flag "Cloud buffer" is applied for a more conservative treatment of clouds. Within the processing other quality criteria are implemented e.g. for cases of failures in the 1Dvar or the PCR.

### 2.2.2 Band characterization OLCI-FLEX

The 45 FLEX like bands are distributed in the visible spectral range between 500 and 792 nm. The oxygen absorption bands have a high coverage with bands every 1.25 nm. The bandwidths are limited by the elementary spectral band of OLCI which is nominally 1.7 nm wide. The other bands are distributed over the spectral range with widths up to 3.7 nm by combining





elementary bands. The nominal central wavelengths and bandwidths of OLCI-FLEX bands are listed in Table 1.

The L0-to-L1b-processing of the 45 OLCI-FLEX bands is based on the regular OLCI OL1 processor, which expects 21 input bands. Thus, the 45 bands were split in three subsets with 21 bands each (called FX1, FX2, FX3)(European Space Agency, 120 2021; Deru and Bourg, 2019). Each subset covers the visible wavelength range between 500 and 792 nm with equally distributed sample points to achieve a best possible stray light correction during L1b processing.

For the transfer functions, the 21 bands of the set FX1 are the basis of the used data set. Bands which are not part of FX1 data sets are used from the data sets FX2 and FX3. The radiances of duplicated bands are very similar. Thus, the transfer function is only applied to one selection of bands.

### 2.2.3 Band characterization OLCI-A

For the comparison with OLCI-FLEX, 12 OLCI-A bands in the same spectral range have been selected namely Oa05-Oa16. Their nominal band widths vary between 2.5 and 15 nm. Their nominal central wavelengths and band widths are given in Tab. 1.

### 2.2.4 Aerosol information

The AOD is taken from the closest AERONET station to each pixel. For each scene, all stations with valid measurements 130 are selected and the distance to each pixel is calculated with the great circle distance measure. The mean over one day of the measured AOD from the closest station is used as fixed aerosol prior knowledge. The spectral resolution of the AERONET measurements is low. Usually there are only three sample points in the visible range with measurements at 500, 675 and 870 nm. We assume that the AOD varies with $\lambda^{-1}$. To calculate the AOD at the central wavelength of OLCI-FLEX pixel,the 135 AOD is fitted with a $\lambda^{-1}$-function. With this method, the measured spectral extinction of the present aerosol is considered in the transfer function. Other aerosol parameters like the single scattering albedo (SSA) and the phase function are not adapted in the transfer function. Instead, a fixed aerosol model is used in the radiative transfer function (see Section 2.5.3). The precise knowledge about those aerosol properties is not necessary. The effect of this simplification is discussed in Section 4.1.

### 2.2.5 Surface information

Information about surface pressure and surface type are needed for the simulation of TOA radiances. The sea level pressure and the altitude are given in the Sentinel-3 data sets. Using the barometric height formula, the surface pressure is approximated. For a linear temperature gradient of 0.65 K temperature decrease per 100 m the surface pressure p is

$$p = p_0 \cdot \left(1 - \frac{0.0065 \cdot h}{T}\right)^{5.2555}, \tag{1}$$

with $p_0$ sea level pressure in hPa, $T$ temperature at surface in Kelvin and $h$ altitude in meter.

The first guess of the surface reflection for the 1Dvar is selected based on the normalized difference vegetation index (NDVI).



**Table 1.** Overview of band distribution, nominal central wavelength and bandwidth of FLORIS, OLCI-FLEX and OLCI-A.

| FLORIS | | | OLCI-FLEX | | OLCI-A | |
|---|---|---|---|---|---|---|
| **Band** | **FWHM** | **Spectr. sampl.** | **Central wvl** | **FWHM** | **Central wvl** | **FWHM** |
| | | | 500.625, 531.875, 535.625 | 3.7 | 510 | 10 |
| | | | 538.125 | 1.7 | | |
| 500-600 | 3 | 2 | 540.625 | 3.7 | | |
| | | | 543.125 | 1.7 | | |
| | | | 545.625, 550.625, 570.625, 585.625 | 3.7 | 560 | 10 |
| | | | | | 620 | 10 |
| 600-677 | 3 | 2 | 600.625, 615.625, 620.625 | 3.7 | 665 | 10 |
| | | | | | 673.75 | 7.5 |
| 677-686 | 0.6 | 0.5 | 681.87, 683.125, 684.375, 685.625 | 1.7 | 681.25 | 7.5 |
| 686-697 | 0.3 | 0.1 | 686.875, 688.125, 689.375 | 1.7 | | |
| | | | 696.875 | 1.9 | | |
| 697-740 | 2 | 0.65 | 706.875, 710.625, 721.875, 734.375 | 1.7 | 708.75 | 10 |
| 740-759 | 0.7 | 0.5 | 746.875, 755.625 ,756.875, 758.125 | 1.7 | 753.75 | 7.5 |
| | | | 759.375 , 760.625, 761.875,763.125, 764.375 | 1.7 | 761.25 | 2.5 |
| 759-769 | 0.3 | 0.1 | 765.625 | 2.0 | 764.375 | 3.75 |
| | | | 766.875, 768.125 | 1.7 | 767.5 | 2.5 |
| | | | 769.375,770.62,771.875, | 1.7 | | |
| 769-780 | 0.7 | 0.5 | 773.125, 774.375, 775.625, 776.875 | 1.7 | 778.75 | 15 |
| | | | 791.875 | 3.7 | | |

The NDVI is calculated using L1B TOA radiances with

$$NDVI = \frac{I_{791} - I_{681}}{I_{791} + I_{681}}. \tag{2}$$

Based on the NDVI, the surface is classified in surface types as shown in Table 2. For each surface type a surface reflectance spectrum is chosen randomly from the data bases measured by Advanced Spaceborne Thermal Emission Reflection Radiometer
ASTER (Baldridge et al., 2009) or United States Geological Survey USGS (Clark et al., 2007).

## 2.3 Preprocessing

The preprocessing includes the georeferencing to find matching pixels, the gas correction of the TOA radiance, the normalization of the radiance using the inband solar irradiance and the application of the temporal evolution model of the central wavelength developed by Preusker (2021).



**Table 2.** Classification of surface based on NDVI and randomly chosen surface spectra.

| NDVI | Surface Type | Source |
|:---:|:---:|:---:|
| <0 | Flagged out | - |
| 0-0.1 | Soil (Halloysite) | USGS speclib06a |
| 0.1-0.2 | Dry grass | ASTER |
| 0.2-0.3 | Range land | USGS speclib06a |
| >0.3 | Deciduous Forest | ASTER |

### 2.3.1 Georeferencing

For the georeferencing we used the same method that was suggested by Lamquin et al. (2020). They showed that the reprojection of both OLCI-A and OLCI-B on the same regular grid results in a valid georeferencing of OLCI-A and B for the tandem phase data. We reprojected OLCI-A and OLCI-FLEX data on the same regular grid with a resolution of $0.01°$ on the basis of their high-resolution longitude and latitude position taken from the $geo\_coordinates.nc$ files.

### 2.3.2 Gas correction

The gas concentrations of water vapour and ozone are provided in the OLCI L1b data set. The data originate from forecasts of the European Centre for Medium-Range Weather Forecasts (ECMWF). The gas corrected TOA radiance $I_{gas_{corr}}$ can be calculated by scaling the measured TOA radiance $I_{meas}$ with the gas transmission:

$$I_{gas_{corr}} = \frac{I_{meas}}{exp(-c \cdot \tau(cwvl, fwhm) \cdot amf)}. \tag{3}$$

The gas optical thickness $\tau$ is calculated in the k-binning model. The spectral high resolution output of the k-binning model is convolved with the spectral response functions of OLCI-FLEX and OLCI-A, which results in a central wavelength (cwvl) and FWHM dependence of $\tau$. The scaling factor $c$ is the ratio of the provided gas concentration and the gas concentration used for the k-binning model. The path length of the light is approximated by the air mass factor $amf$. It is calculated using the sun zenith angle (SZA), viewing zenith angle (VZA):

$$amf = \frac{1}{cos(SZA)} + \frac{1}{cos(VZA)}. \tag{4}$$

The gas correction in this simplified way is possible since the interaction of absorption and scattering is weak.

### 2.3.3 Time-evolution of band characteristics

OLCI's spectral characteristics are regularly monitored in-flight using spectral campaigns. The procedures use the programming capability of OLCI to define 45 bands around stable spectral features, to characterize the spectral dispersion of each camera system with respect to the spectral and the spatial (across track) dimension. Simulations of OLCI measurements in



the 45 bands are optimized for best agreement with the spectral features, as a function of assumed bandwidth and band center wavelength of an individual CCD element. Depending on the used spectral feature the achieved accuracy for the central wavelength is in the order of 0.1-0.2 nm, the precision (repeatability) is better than 0.05 nm. The regularity of the spectral campaigns allows a precise quantification of the temporal evolution of the spectral response for each individual CCD-element on
each camera CCD, at least for the investigated spectral features. It emerges that all cameras show a tiny but distinct evolution. Both, the single CCD-row bandwidth and the across track variability ('smile') of the central wavelength remain almost constant for all cameras of OLCI A and B, but the central wavelengths of all pixels move almost homogenously with a decreasing rate. Since launch, four of the five cameras of OLCI A and B respectively, have drifted up to 0.3 nm towards longer wavelengths. One camera (camera 5 for OLCI A and camera 3 for OLCI B) has drifted by 0.3 nm to shorter wavelengths. The dependency
of the central wavelength on the orbit can be described with the following model (Preusker, 2021):

$$cwl = a + b * ln(orbit) + c * ln(orbit)^2. \tag{5}$$

The coefficients a, b and c are published for OLCI-A and OLCI-B band sets for any band, pixel and orbit in https://sentinel.esa.int/documents/247904/2700436/LUT.zip. With those coefficients, the central wavelength can be calculated for an arbitrary orbit of OLCI-A or -B.
The temporal evolution of the OLCI-FLEX spectral characterization is based on the spectral shift of the OLCI-B band Oa12 at 753.75 nm (nominal). Its temporal shift is applied for all OLCI-FLEX bands within the $O_2A$ absorption band. This approach is valid because of the homogeneous behaviour of the temporal evolution across the spectrum.

The central wavelengths of OLCI-A are also shifted using the described model and the corresponding LUTs.

## 2.4    Radiative Transfer Simulations

The radiative transfer simulations used to build LUTs for the transfer function were computed using the radiative transfer model "Matrix Operator Model" (MOMO) developed at Freie Universität Berlin (Hollstein and Fischer, 2012; Fell and Fischer, 2001). It is a doubling and adding model based on a layered description of a plane-parallel atmosphere which can be coupled with an ocean-optical model or any surface bidirectional reflectance function. It solves the matrix form of the radiative transfer equation after discretizing it. Scattering functions of aerosol particles are calculated with the Mie algorithm (Wiscombe, 1980).
Gas absorption is implemented using a k-binning description of line-by-line models (Doppler et al., 2014).

## 2.5    Radiative Transfer Input

### 2.5.1    Atmospheric Profile

The simulated atmosphere is divided in plane-parallel layers. Molecules and particles are distributed homogeneously within each layer. The vertical distribution of the atmospheric gases is based on a standard vertical temperature, pressure and humidity
profile defined by AFGL Atmospheric Constituent Profiles (Anderson et al., 1986).

For the simulation of the OLCI-FLEX and OLCI-A measurements the *mid-latitude summer* profile was used as all acquisition





sites are in Europe and the campaign took place from 14th June till 14th August 2018. The standard profile is interpolated to build a model of the atmosphere that contains up to 23 layers with level borders every 50 hPa. The surface pressure is either 700, 800, 900, 1013 and 1050 hPa. Lower surface pressures reduce the number of levels.

### 2.5.2 Gas Absorption

For the description of the wavelength dependent gas absorption processes, we use HITRAN16 database (Gordon et al., 2017). For the oxygen ($O_2$) absorption we use the cross sections of Drouin et al. (2017). All relevant atmospheric gases except of ozone and water vapour are considered in the simulations. Due to the weak interaction of absorption and scattering in the considered bands, the effect of those gases on the TOA radiance can be corrected by a simple transmission correction (see 2.3.2). The interaction between absorption of $O_2$ and scattering is strong in the oxygen absorption bands. Thus, TOA radiances must be calculated for different $O_2$ amounts. $O_2$ is a well mixed gas in the atmosphere and it scales with the surface pressure. The MOMO simulations are done for atmospheric profiles with different surface pressures to consider the effect of the $O_2$ interactions.

### 2.5.3 Aerosol Model

A continental aerosol model from the OPAC database (Hess et al., 1998) is used for the simulations. The aerosol particles contain insoluble particle, water soluble particles and soot. The refractive index and size distribution are given for a relative humidity of 80%. The size distribution of each component is a log-normal distribution. Its coefficients and their refractive indices are given in Table 3. From those parameters the extinction coefficient and SSA are calculated. The wavelength dependence of the extinction coefficient is typical for a continental aerosol (Fig. 3a). Using Mie-scattering theory, the phase function is developed for 171 different scattering angles between 0 and 180° (Fig. 3b). The spectral dependence of the phase function $p(cos\Theta)$ can be described with the asymmetry parameter $g$:

$$g(\lambda) = \frac{1}{2} \int\limits_{-1}^{1} p(cos\Theta) dcos\Theta \tag{6}$$

For the radiative transfer simulations, the aerosol particles are placed homogeneously distributed in the layer closest to the surface. All simulations are done with a reference AOD at 550 nm ranging from 0.05 to 0.8.

Using one aerosol model is possible as our method does not require a perfect description about the atmosphere. The effect of this simplification is discussed in section 4.1.

### 2.5.4 Surface reflectance

The spatial resolution of OLCI is about 300 x 300 m at nadir. The covered surface is most likely a mixture of different surface types in our study areas in Europe. It is simplified as isotropic reflector with surface reflectances between 0.01 and 0.81. An isotropic reflector reflects the light in all directions with the same probability. In reality, most surfaces have a distinct angular dependent probability function for the reflection of light.





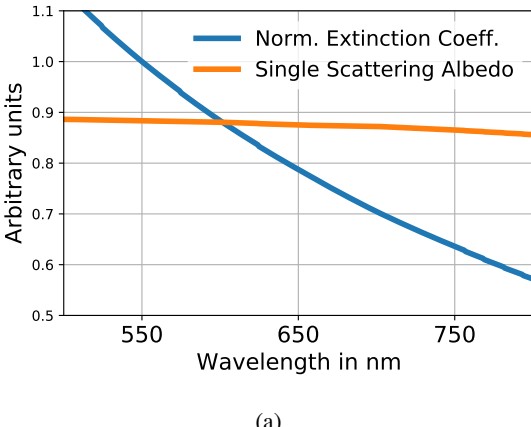
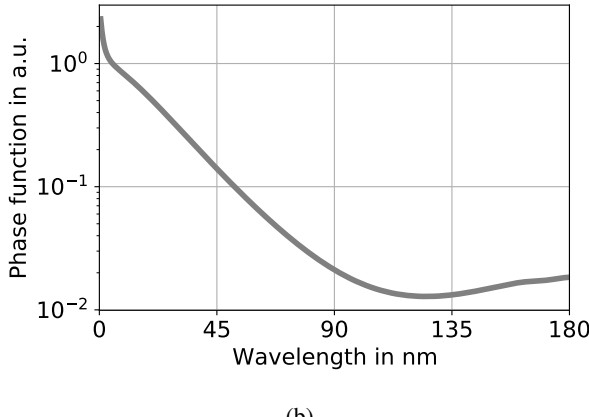

(a)                                              (b)

**Figure 3.** (a) Wavelength dependence of normalized extinction coefficient (blue) and SSA (orange) for the defined continental aerosol for reference wavelength 550 nm. (b) Angle dependent phase function at 550 nm for same continental aerosol.

**Table 3.** Information about aerosol components, refractive indices are given at 550 nm.

|  | Insoluble | Water soluble | Soot |
|---|---|---|---|
| **Size Distribution Coeff. a** | 0.471 | 0.306 | 0.0118 |
| **Size Distribution Coeff. b** | 2.51 | 2.24 | 2.00 |
| **Refractive index (real)** | 1.53 | 1.399 | 1.75 |
| **Refractive index (imag.)** | -0.8E-02 | -0.199E-02 | -0.44 |

## 2.6 Radiative Transfer Output

The output of MOMO is the diffuse up- and downwelling radiance at each layer for the simulated atmosphere and each representing rectangular band. The radiances are sun and viewing angle dependent. For each k-term the radiance and a weight is calculated. The radiances have the unit $sr^{-1}$ for an associated solar constant of one.

The simulated upwelling radiances are convolved with the spectral response functions of OLCI- A (Sentinel 3 CalVal Team, 2016) and of OLCI-FLEX. The spectral response functions of OLCI-FLEX are approximated with Gaussian functions. This approximation does not hold for OLCI-A spectral response functions. Thus, the actual response function shapes are used to convolve the simulations for the OLCI-A LUTs.

## 2.7 1D-variational approach

The information needed to describe the atmospheric state are determined by an 1D-variational approach (1Dvar). It finds the most probable state that describes the radiance measurement starting from *apriori* knowledge. The approach is implemented following Rodgers (2000).





**Table 4.** Dimensions of LUTs.

|  | **Surface refl.** | **Surface pr.** | $AOD_{550}$ | **Central wvl** | **FWHM** | **SZA** | **VZA** | **ADA** |
|---|---|---|---|---|---|---|---|---|
| **Minimum** | 0.01 | 700 hPa | 0.055 | Band$-0.5$ nm | Min FWHM | $0.0^o$ | $0.0^o$ | $0.0^o$ |
| **Maximum** | 0.81 | 1050 hPa | 0.94 | Band$+0.5$ nm | Max FWHM | $88.49^o$ | $88.49^o$ | $180.0^o$ |

The 1Dvar is an iterative process comparing a forward simulated radiance $\boldsymbol{F(X)}$ with the measured radiance $\boldsymbol{Y}$ (with capital letter referring to vector and matrices):

$$\boldsymbol{G(X_i) = F(X_i) - Y} \tag{7}$$

The state vector $\boldsymbol{X_i}$ is adjusted in each step i using the Gauss-Newton method:

$$\boldsymbol{X_{i+1} = X_i - (S_a}^{-1} + \mathbf{K_i}^T \mathbf{S_e}^{-1} \mathbf{K_i})^{-1} (\mathbf{K_i}^T \mathbf{S_e}^{-1} \cdot \boldsymbol{G(X_i)}$$
$$- \mathbf{S_a}^{-1} \cdot \boldsymbol{(X_a - X_i))} \tag{8}$$

The difference between forward model and measurement is weighted with the measurement error co-variance matrix $\mathbf{S_e}$ and the Jacobian $\mathbf{K_i}$. Furthermore, the difference between parameter state vector and *apriori* knowledge $X_a$ is taken into account evaluating also the *apriori* error co-variance matrix $\mathbf{S_a}$. Using the Jacobians, the next step is selected in the direction of the largest gradient. The iteration stops when either the maximum number of iteration (10) is reached or the increment weighted by retrieval error co-variance matrix $\hat{\mathbf{S}}$ is small (Eq. 10). The retrieval error co-variance matrix is given by

$$\hat{\mathbf{S}} = (\mathbf{S_a}^{-1} + \mathbf{K_i}^T \mathbf{S_e}^{-1} \mathbf{K_i})^{-1}. \tag{9}$$

It is weighted with the step width between two following states, giving the stop criterion:

$$\boldsymbol{(X_i - X_{i+1})}^T \cdot \hat{\mathbf{S_i}}^{-1} \cdot \boldsymbol{(X_i - X_{i+1})} > n \cdot \epsilon \tag{10}$$

$\epsilon = 0.01$ and $n$ is the number of parameter state dimensions. This method can be applied under the assumption of Gaussian probability density functions of uncertainty and bias-free measurements, priors and models.

### 2.7.1 Look-up tables and interpolation

The parameters used in the 1Dvar and the forward model are the surface reflectance, surface pressure, the AOD, central wavelength, FWHM, SZA, VZA and azimuth difference angle (ADA). Simulations with variations of those parameters are stored in LUTs. Their dimensions are summarized in Table 4. The parameter dimensions of the LUTs are regularly spaced, allowing a fast indexing and interpolation for the forward operator. The step width for the central wavelength and the FWHM is 0.1 nm. The N-dimensional interpolation of $\boldsymbol{X^*}$ in a regular parameter space $[p_1, p_2, ..., p_n]$ is divided into the following two steps:



1. Normalization of the input variables:

$$p_i^* = \frac{p_i - p_i^l}{p_i^u - p_i^l}$$ (11)

where $pi^u$ and $p_i^l$ is the nearest lower and the nearest upper parameter entry in the LUT.

2. Interpolation by a weighted sum of the $2^N$ enveloping neighbours in the LUT:

$$\begin{aligned}
\boldsymbol{X^*}(p_1,...,p_n) = {} & (1-p_1^*)(1-p_2^*)...(1-p_n^*)\boldsymbol{X}^{l,l,...,l} \\
& + (0-p_1^*)(1-p_2^*)...(1-p_n^*)\boldsymbol{X}^{u,l,...,l} \\
& + ... \\
& + (0-p_1^*)(0-p_2^*)...(0-p_n^*)\boldsymbol{X}^{u,u,...,u}
\end{aligned}$$ (12)

### 2.7.2 Optimization of OLCI-FLEX radiances

The 1Dvar is applied to the gas corrected OLCI-FLEX radiance measurement to find best possible characterization of the atmosphere and the surface. The atmosphere is parameterized with the surface pressure from the L1b data of OLCI-A and the standard vertical profile used in the radiative transfer simulations (see Sect. 2.5.1). The surface reflectance is optimized pixelwise and bandwise using the 1Dvar approach. A randomly selected surface reflectance spectrum according to the classified surface is used as *apriori* knowledge. This approximation is very rough and though it is handled with a large *apriori* error of 1. All other state parameters, namely band characterization (central wavelength and FWHM), wavelength dependent AOD, surface pressure, and measurement geometry (SZA,VZA, ADA), are kept constant and are taken from the sources described in Section 2.2. The measurement error co-variance contains the signal-to-noise ratio of OLCI which is approximately 1:200. The retrieved spectral dependent surface reflectance is used in the next step of the transfer function as input of the PCR, which is described in the next section.

### 2.8 Spectral interpolation of surface reflectance

OLCI-FLEX bands do not cover all spectral features of the surface reflectance needed to reconstruct the surface reflectance at the lower resolution OLCI-A. An interpolation of the surface reflectance at nominal OLCI-A bands at 510, 560, 665, 673.75 and 681.25 nm is not possible due to the gaps in the OLCI-FLEX spectrum (see Tab. 1). We decided to use a principal component regression (PCR) to fill the missing gaps. A set of high resolution surface reflectance spectra from the spectral libraries USGS (Clark et al., 2007) and ASTER (Baldridge et al., 2009) are decomposed into eigenvectors. Depending on the NDVI of each pixel the data base for the PCR is chosen. For a low NDVI (<0.2) all spectra in the USGS soil data base are used, the range land spectra from USGS vegetation data base are used for pixels with NDVI between 0.2 and 0.3 and all vegetation spectra except of the range land spectra from USGS plus the grass and forest spectra from the ASTER data base are used for pixels with high NDVIs (>0.3). The found eigenvectors are called principal components since a linear combination of those can





reconstruct an arbitrary surface reflectance spectrum. The decomposition into eigenvectors and the linear regression to find the linear coefficients are made using the Python library scikit-learn (Pedregosa et al., 2011). For each pixel a set of 4 and a set of 6 principal components are found. The set with the minimum mean squared error between reconstructed and input surface reflectance is chosen. If no valid reconstruction can be found, the pixel is flagged. Similarly, Vidot and Borbas (2014) found that six principal components is the optimal input to reconstruct hyperspectral surface reflectance spectra from seven MODIS

bands.

The PCR is used to transfer the surface reflectance retrieved at OLCI-FLEX bands to OLCI-A bands Oa05 (510 nm), Oa06 (560 nm), Oa08 (665 nm), Oa09 (673.75 nm) and Oa10 (681.25 nm). The other OLCI-A bands are transferred by a linear interpolation of the OLCI-FLEX bands.

## 2.9 Forward simulation

The found information of the surface and the atmosphere serve as input for the forward model that reconstructs OLCI-A TOA radiance measurements. All wavelength dependant information are shifted from the OLCI-FLEX bands to the OLCI-A bands. A combination of PCR and linear interpolation shifts the surface reflectance to the OLCI-A bands. The AOD originating from AERONET is also interpolated. The mean surface pressure from the 1Dvar, band characteristics of OLCI-A (central wavelength, FWHM) and measuring geometry serve as input for the forward model without further transformation. The output

of the forward model is a TOA radiance at the OLCI-A bands that is based on information gained from OLCI-FLEX and thus, which is representative for the OLCI-FLEX measurement. The forward model is applied band and pixelwise.

## 3 Results

### 3.1 Transfer Function applied on Single Pixel

We applied the transfer function pixelwise. The results for an example pixel west of Paris from the 2nd July 2018 is shown in

Fig. 4. We chose the pixel due to the good agreement of the measured radiances at the first glance and the small spatial distance of 140.5 m between the pixel centers of OLCI-FLEX and OLCI-A. The good agreement indicates that the measurements are not affected by an heterogeneous surface or atmosphere. In the upper left plot, the gas-corrected TOA radiances are given. The spectral distribution of the bands shows the discussed gaps of OLCI-FLEX bands between 500 and 520 and between 650 and 680 nm. The OLCI-AR radiances, created by applying the transfer function on OLCI-FLEX, differ slightly from the measured

radiance OLCI-A. Only in the last band OLCI-A and OLCI-AR and OLCI-FLEX radiances deviate more. We use the relative difference between OLCI-A and the OLCI-AR to quantify the agreement between the two data sets.

$$\Delta I = \frac{I_{OLCI-AR} - I_{OLCI-A}}{I_{OLCI-A}} * 100. \tag{13}$$

The difference is shown in the lower left subplot of Figure 4. The negative difference between the OLCI-AR and OLCI-A data indicates that the radiance measured by OLCI-A is brighter than the one measured by OLCI-FLEX. Only at longer wavelength

(780 nm) OLCI-FLEX is darker. The relative difference between 550 and 680 nm is approximately one percent. Between




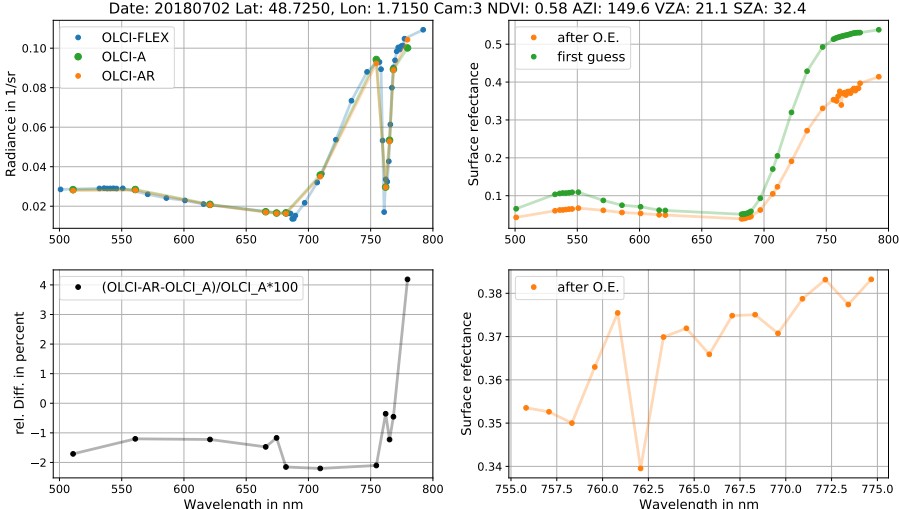

**Figure 4.** Results of transfer function for one reprojected pixel on the 2nd July 2018. Upper left: gas corrected measurements and OLCI-AR; upper right: first guess of the surface reflectance and the one from the 1Dvar approach (after optimal estimation OE); lower left: relative difference between reconstructed and measured OLCI-A radiance, lower right: optimized surface reflectance in $O_2A$ absorption band.

681.25 and 753.75 nm, we can observe a decrease of the relative difference to two percent and a strong gradient for the bands within and behind the $O_2$ absorption band.

The information about the surface that describes the state observed by OLCI-FLEX is given in the right subplots of Figure 4. The surface reflectance has a strong red-edge and a smooth spectrum as typical for vegetated surfaces. Only in the oxygen
band the surface reflectance shows small oscillations (see lower right plot in Fig. 4). Reasons for oscillations in $O_2$ band can be errors in spectral characterization of OLCI-FLEX and OLCI-A, in the surface pressure and the instrument measurement uncertainty. Most probably it is a combination of all three reasons, which are not entangled in the scope of this paper.

## 3.2 Statistical Evaluation of Relative Difference between Satellites

A large range of relative differences between OLCI-AR and OLCI-A radiances can be observed when studying different pixels. It depends on the validity of the geo-referencing and the quality of the PCR. To reduce the effects of these uncertainties, a statistical evaluation of many pixels is necessary. The median value of the relative difference is calculated for different parameters for land pixels of the OLCI-FLEX scene on the 2nd July 2018. The median is chosen because it is less sensitive to outliers. The area covered in this scene is shown with the red frame in Fig. 2.





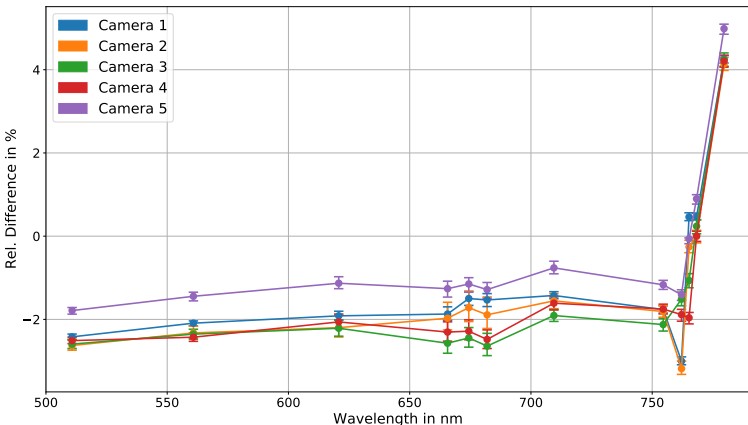

**Figure 5.** Median relative difference between OLCI-AR and OLCI-A radiance for each camera. All valid pixel of 2nd July 2018 were used, which gives about 200 000 pixels per camera.

### 3.2.1  Median by camera

The first statistical evaluation is done by taking the median of all pixels for each camera in our study scene. Each camera data set contains more than 200 000 measurements. Figure 5 shows the median relative difference for each camera for all valid land pixels measured by OLCI-FLEX and OLCI-A. The median is negative with values of about 2%. Only camera 5 shows a smaller relative difference of about 1%. These results confirm that overall OLCI-FLEX measures darker radiances than OLCI-A. Only at the 780 nm the difference reaches positive values of more than 4%.

To quantify the representativity of the median we use a bootstrap method: 1 000 random subsets of about 20 000 pixels were selected to calculate the median. The minimum and the maximum median value serve as lower and upper error bound in the plot. The resulting error bars at 680 nm are slightly larger which is due to the lack of knowledge about the surface reflectance at those bands. This gap is filled with the PCR which introduces an uncertainty.

The medians for the cameras allow a visualisation of the wavelength dependent difference. We can observe a slight increase with the wavelength between 500 and 750 nm. Small features within the oxygen band at 760 nm ($O_2A$ band) indicate errors in the description of the band characterization or the atmospheric parameters. The strong gradient in the relative difference behind the $O_2A$ band is not expected.

In Figure 5, we observe that camera 3 and camera 5 show almost no absorption feature of the $O_2A$ band. Whereas camera 1 and 2 have the strongest absorption features. This observation is even more striking in Fig. 7. From that observation, we conclude that the assumed spectral response functions are very accurate for camera 3 and 5. Reasons for the absorption features in camera 1,2 and 4 are either the spectral characterization or the occurrence of aerosol types which are not represented by the simulated aerosol model. The effect of a wrong aerosol model is discussed in Section 4.1.





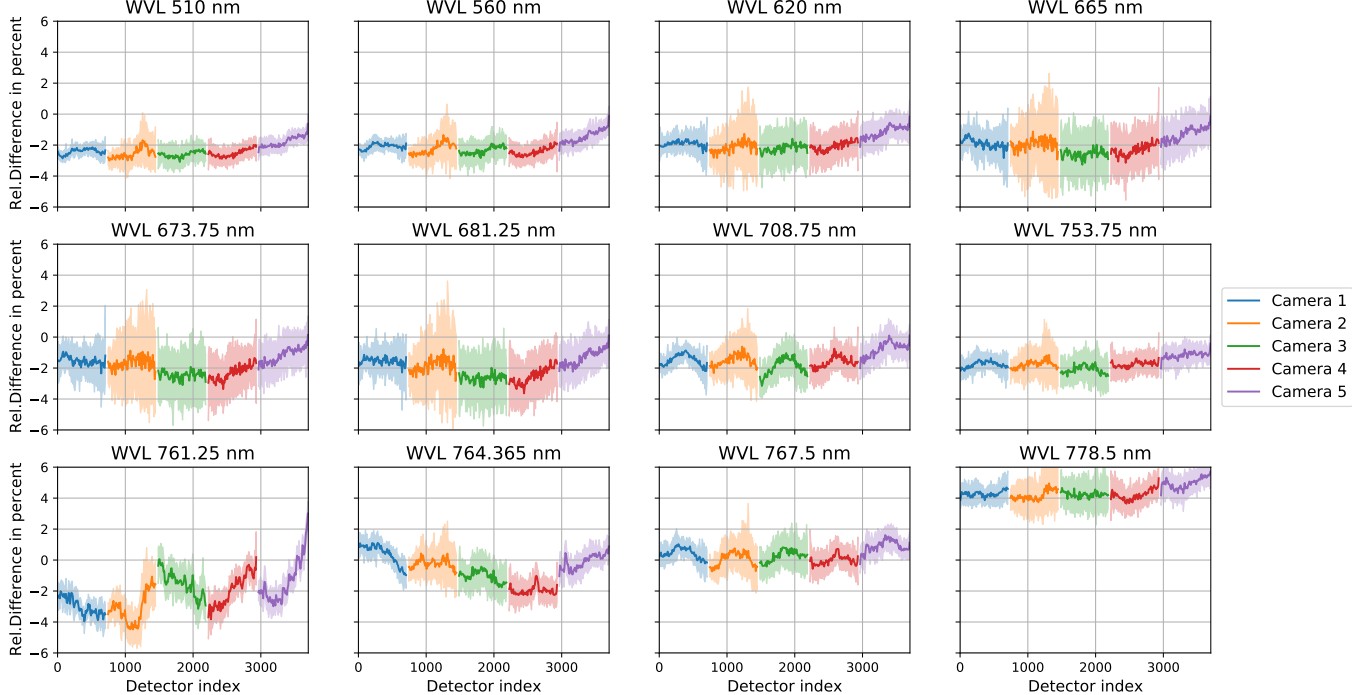

**Figure 6.** Median over detectors of relative difference between OLCI-AR and OLCI-A radiance. Every 10 detector indices were binned together, each bin has at least 500 entries. Each subplot gives one OLCI-A channel. The 740 detectors of each camera are serially numbered from west to east: detector indices 0 till 740 belong to camera 1, 741 till 1480 to camera 2,... The different cameras are colour coded. Shaded area shows representativity.

### 3.2.2 Median by Detector Index

The camera effects are studied in more detail with the median of the relative difference for each detector which is presented in Fig. 6. Every 10 detectors are binned together. The median is only taken for bins with more than 1000 entries. The representativity is shown in the shaded areas. It is estimated with the bootstrap method described before. In 100 iterations the median was calculated for subsets of one tenth of data points. The minimum and maximum median are the borders of the shaded area.
The variation in the relative difference across the field of view is strongest for camera 5. The drift of relative difference in

camera 5 explains the difference of the median over the camera compared to the other cameras seen in Fig. 5.
Further camera effects can be observed for all cameras at 708.75 nm. This band is influenced by water vapour absorption. The relative difference between OLCI-AR and OLCI-A radiances are larger at the camera edges. The same effect can be observed at 767.5 nm which lies within the weaker part of the oxygen absorption band. The bands at 761.25 and 764.375 nm are in the spectral area with sharp oxygen absorption lines. Here, the largest variations in the median can be observed.

The largest uncertainty shows camera 2 especially in the nominal bands 665, 673.75 and 681.25 nm, which confirms the observations in Fig. 5.




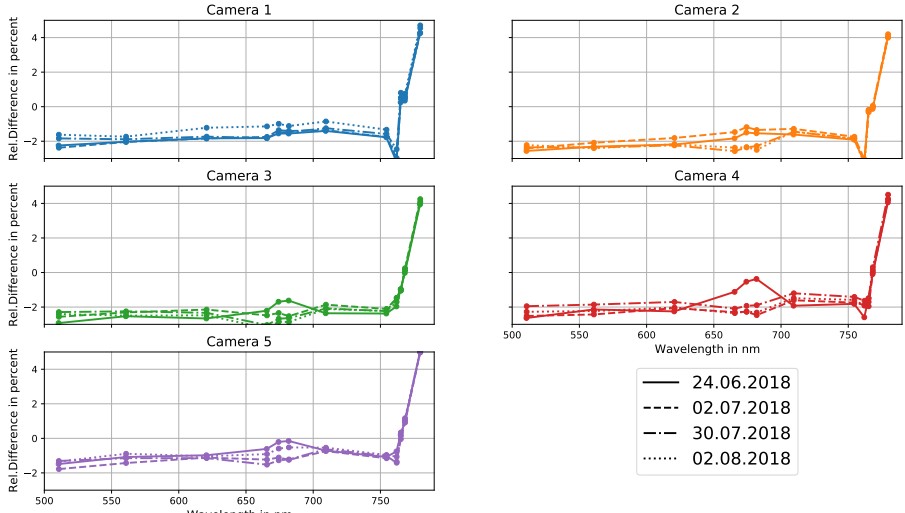

**Figure 7.** Median of rel. Difference between OLCI-AR and OLCI-A radiances over all valid of scenes at selected days of the campaign.

Overall, the relative difference is about 2% over most detectors and wavelengths. Only at the end of the studied spectrum, the relative difference becomes positive with values of up to 5%. The results of the median for the individual detectors agree with camera median.

### 3.2.3 Transfer Function applied on Time Series

The tandem phase of OLCI-FLEX and OLCI-A lasted three months from May 2018 till August 2018. All scenes were recorded over a similar part of Europe (see Fig. 2). Nevertheless, the underlying surface changed over time as the tandem phase was during the crop harvesting season. Additionally, observations of OLCI-A showed that the instrument is aging most strongly during the first months after launch. The tandem phase of OLCI-FLEX and OLCI-A was during the commissioning phase of OLCI-B just after its launch. Hence, a time-dependent study of the difference between OLCI-FLEX and OLCI-A is necessary. The transfer function was applied on scenes at the beginning, in the middle and the end of the mission and for scenes with a small cloud coverage.

Figure 7 shows the comparison of the median relative differences over camera for four selected days. For all four days the spectral shape of the relative difference is similar for all cameras. The largest deviations among the different days are between 660 and 680 nm. As discussed before, it is the spectral region in which the method shows its largest uncertainty due to the lack of bands in the OLCI-FLEX setting. Within the spectral region the validity of the transfer function depends on the quality of the PCR which differs from pixel to pixel. This feature is most prominent on the 24th of June especially in camera 4. However, we cannot observe a systematic time dependence of the relative difference. Hence, the effect is due to coincidence of other differences among the scenes, e.g. covered area or cloud coverage. A small difference between all days and the 2nd of August can be observed in camera 1. However, the time step to the next study scene is only three days. Within such short period we do



not expect such a change in the camera characterization.

Overall, the deviation in relative difference shows no systematic features in the different cameras over time. Hence, the difference between OLCI-FLEX and OLCI-A has no significant time dependency within the studied time period. This result shows the quality of used time-evolution model of the band characterization. The wavelength does change over time by maximum

0.1 nm, if this change was not considered, we would have seen a time dependence in the relative difference.

## 4    Sources of uncertainty

The sources of uncertainty affecting the method of the transfer function are discussed qualitatively in the following section. Besides the uncertainty introduced by simplifications and assumptions of the radiative transfer model, uncertainty is primary introduced by the uncertainty of the input data for the inversion and the forward model, namely the radiance, surface pressure,

measurement geometry, spectral response, the total column water vapour, ozone concentration, co-registration, and aerosol parameters. Most data are used from the level 1 files of OLCI-A (Section 2.2).

The uncertainty of radiance is estimated with a signal to noise ratio of 200 and it is considered in the diagonal elements of the measurement co-variance matrix. We assume no co-variances. The uncertainty of measurement geometry, the band characterization, water vapour and ozone content are not propagated within the transfer function. Nevertheless, the effect of

band characterization is qualitatively discussed in Section 4.3.

The systematic uncertainty introduced by errors in the misalignment are already discussed in Section 3.2.1. By taking the median over a large data set the uncertainty due to misalignment is minimized. The good representativity of the median is shown by the small error bars gained in the bootstrap method. It implies an accurate determination of the bias between OLCI-A and OLCI-FLEX.

The effect of the fixed aerosol model and the rough assignment of AERONET data to the pixels and the coupled errors in AOD are discussed in the next section. We also show their impact on the quality of the surface reflectance in Section 4.2. Additionally, the effect of the PCR is discussed briefly.

### 4.1    Aerosol-model sensitivity

Two simplifications in the description of the aerosol can induce uncertainties. Firstly, we fix the phase function and the SSA

according to a single aerosol model. Secondly, the AOD and the spectral extinction of the aerosol is approximated with data from the closest AERONET station. The following sensitivity study shows the effect of the fixed aerosol phase function and SSA and a wrong spectral AOD on the transfer function and on the reconstructed spectra. Simulated high-resolution spectra with different aerosol models serve as input of the transfer function which is based on fixed aerosol model. The output of the transfer function, the reconstructed spectrum, is compared to the matching simulated low-resolution spectrum. From the

comparison, we can estimate the error induced by the simplification of using a fixed aerosol type. The fixed aerosol model is a continental aerosol as shown and described in Section 2.2.4.





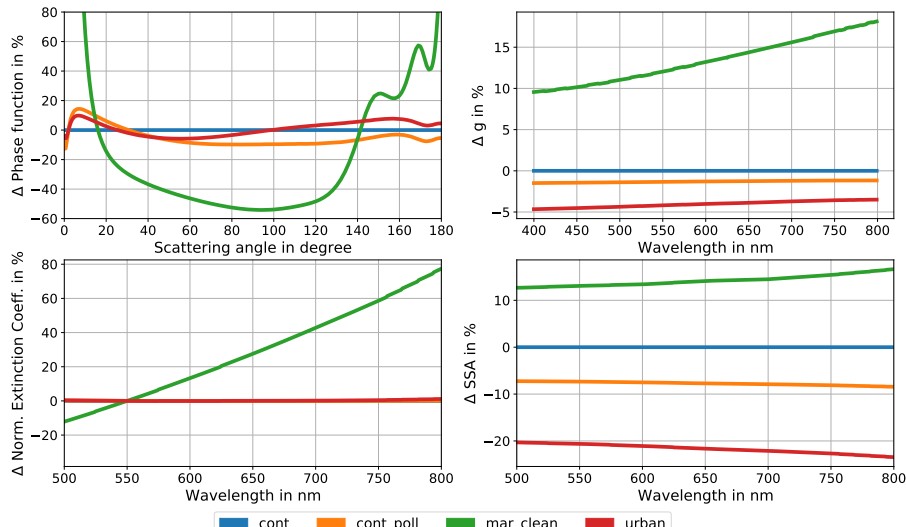

**Figure 8.** Aerosol properties of input aerosol types for sensitivity study. Properties are relative to properties of continental aerosol (see Fig. 3). The different types are continental, continental polluted, maritime clean and urban aerosol as defined in OPAC data bank.

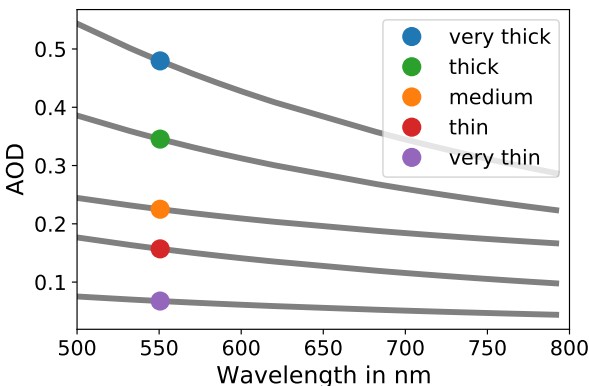

**Figure 9.** The spectral aerosol extinction taken from different AERONET stations. The reference AOD at 550 nm is marked with a coloured circle. Each spectrum will be referred to as optical very thick ($AOD_{550} = 0.48$), thick ($AOD_{550} = 0.35$), medium ($AOD_{550} = 0.23$), thin ($AOD_{550} = 0.16$) and very thin ($AOD_{550} = 0.07$).

### 4.1.1 Experimental set up

We created two scenarios showing the effect of

1) wrong phase functions and SSA but correct spectral extinction of the aerosol.

2) wrong phase functions and SSA and wrong spectral extinction of the aerosol.





Both scenarios are based on the same set of simulated radiances.

As scatterer, we chose four different predefined aerosol models of the OPAC data source (Hess et al., 1998) namely *continental*, *continental polluted*, *urban* and *maritime clean* aerosol. Those models cover the variety of strong and less absorbing aerosols with different phase functions which we would expect in summer over Europe. We assume a vegetated surface and a
surface pressure of 1013 hPA.

In Figure 8 the aerosol properties of the chosen aerosol models are shown relative to the continental aerosol. The maritime clean aerosol differs most strongly from the continental aerosol in three parameters, phase function, asymmetry parameter, and the normalized spectral extinction coefficient. The urban aerosol is strongly absorbing, which can be seen in the relative difference of SSA.

Both scenarios are set up for different optical thicknesses of the aerosol. Five different AOD spectra retrieved at different AERONET stations within Europe on the 2nd of July 2018 were chosen. The spectra are shown in Fig. 9. We forced the chosen aerosol models to follow the measured spectral extinction shown in Fig. 9. As a result, we got 20 aerosol extinction spectra which serve as input for the simulations. For each spectrum we simulated an OLCI-FLEX and OLCI-A radiance spectrum.

We applied the transfer function on the 20 simulated OLCI-FLEX spectra. The LUTs are the same as described in Sections 2.5
and 2.7.1 with the continental aerosol model used. The two scenarios differ in the aerosol input parameters for the inversion of OLCI-FLEX spectra and the forward model. In both scenarios, the *apriori* knowledge about the surface was a surface reflectance spectrum of vegetated land with a slight different spectral shape compared to the truth. The difference between the scenarios is the choice of the aerosol input. In scenario 1, the correct spectral extinction of the aerosol serves as input for the inversion and the forward model. In reality that means, that the measured spectral AOD from the AERONET station represents
the present aerosol but the usage of the continental aerosol model in the LUT induce errors due to its phase function and SSA. The input for scenario 2 is the spectral extinction of the "thin" aerosol with a reference AOD at 550 nm of 0.16. In 16 cases this aerosol description does not represent the actual spectral extinction of the aerosol.

### 4.1.2   Results

The difference between simulated OLCI-A spectra and reconstructed OLCI-A spectra which are based on simulated OLCI-
FLEX spectra are shown in Fig. 10 for both scenarios. It is calculated using Eq. 13. Scenario 1 is plotted in solid lines and scenario 2 in dashed lines. The relative difference between the output of the transfer function and the simulated OLCI-A radiance has the same order of magnitude for all cases.

The cases with the continental aerosol in scenario 1 serve as control case. They are shown in the upper left plot. In those cases we do not insert any errors in the description of the continental aerosol neither in the AOD nor the phase function or the SSA.
Nevertheless, the relative difference is not as expected zero for all bands. The deviation of the relative difference in the intensity shows the residual error made within the transfer function. It originates from the interpolation of the surface reflectance from OLCI-FLEX bands to OLCI-A bands as shown in Fig. 11 and discussed in the next section 4.2.

From the other cases of scenario 1, we can estimate the effect of the difference in the phase function and the SSA (see upper right and lower subplots in Fig. 10). The three cases are based on a different aerosol model but the correct spectral extinction.



The major difference to the control case with the continental aerosol model is within the oxygen absorption band. The results of the three cases with wrong aerosol model show the same deviations in relative difference from zero where the control case shows a relative difference of zero. This deviation shows that the oxygen absorption band is sensitive to the aerosol model, but the choice of aerosol model is not important. The choice of aerosol model effects the bands between 665 and 681.25 nm. Here, the gaps in the band distribution of OLCI-FLEX are filled with additional knowledge from the PCR (see Section 4.2).

The largest absolute relative difference in intensity are shown by the results with the urban aerosol model. This aerosol model is most absorbing and thus its effect on the TOA radiance is strongest. This effect is only visible between 665 and 681.25 nm. For all cases, the relative difference is increased slightly with the AOD for the 665 nm band. The other bands show rather a decrease with the AOD. With an increasing AOD, the TOA radiance is less sensitive to the underlying surface. Thus, the errors in the interpolation affecting the signal less and the relative difference decreases with the AOD. Only between 665 and

681.25 nm the error due to the interpolation increases with the AOD.

Scenario 2 shows very similar results. The largest differences between scenario 1 and 2 are again at between 665 and 681.25 nm. With the optical thickness, the effect of the wrong aerosol model used in the LUTs is increased.

Over all, even for this scenario the relative differences between the reconstructed OLCI-A radiance and the true OLCI-A radiance do not exceed 0.5% in all bands but those which are in the gaps of OLCI-FLEX. Here, only for the cases with a thick

or very thick aerosol the relative difference goes up to 1.2 %. However, In the studied scene less then 1 % of all pixels had a reference AOD at 550 nm of more then 0.3. Hence, the cases of a thick or very thick aerosol layer occur only rarely.

In contrast to the results of the measurements during the tandem phase, we cannot observe a systematic bias over all bands for the cases of our sensitivity studies. Thus, we conclude that the difference between OLCI-AR and OLCI-A shown for the 2nd July 2018 (Fig. 5 and 6) is not an artefact of the transfer function but a systematic difference between OLCI-FLEX and

OLCI-A.

## 4.2 Surface Reflectance Sensitivity

The sensitivity of the description of the surface on the transfer function is studied based on the data simulated for the two scenarios of the aerosol sensitivity study. We studied the optimized surface reflectance found for the 20 cases described above. The surface reflectance retrieved for the OLCI-FLEX bands is interpolated to OLCI-A bands as described in Section 2.8. The

difference of the interpolated surface reflectance to the truth (input for simulations) at OLCI-A bands is shown in Fig. 11. It is calculated with

$$\Delta\alpha = \frac{\alpha_{OLCI_{AR}} - \alpha_{Truth}}{\alpha_{Truth}} * 100. \tag{14}$$

$\alpha$ represents the surface reflectance. The solid lines represent the case, where the aerosol is perfectly known (scenario 1) whereas the dashed lines show the deviation of the true surface reflectance for a retrieval based on a wrong AOD (scenario

2). Additionally, the retrieved OLCI-FLEX surface reflectance is shown for scenario 1 with cross like symbols. Looking at the crosses in the upper left plot of Figure 11, the performance of the surface reflectance retrieval can be assessed. The surface reflectance was optimized under perfect conditions. Perfect conditions mean, that we know the correct spectral response func-





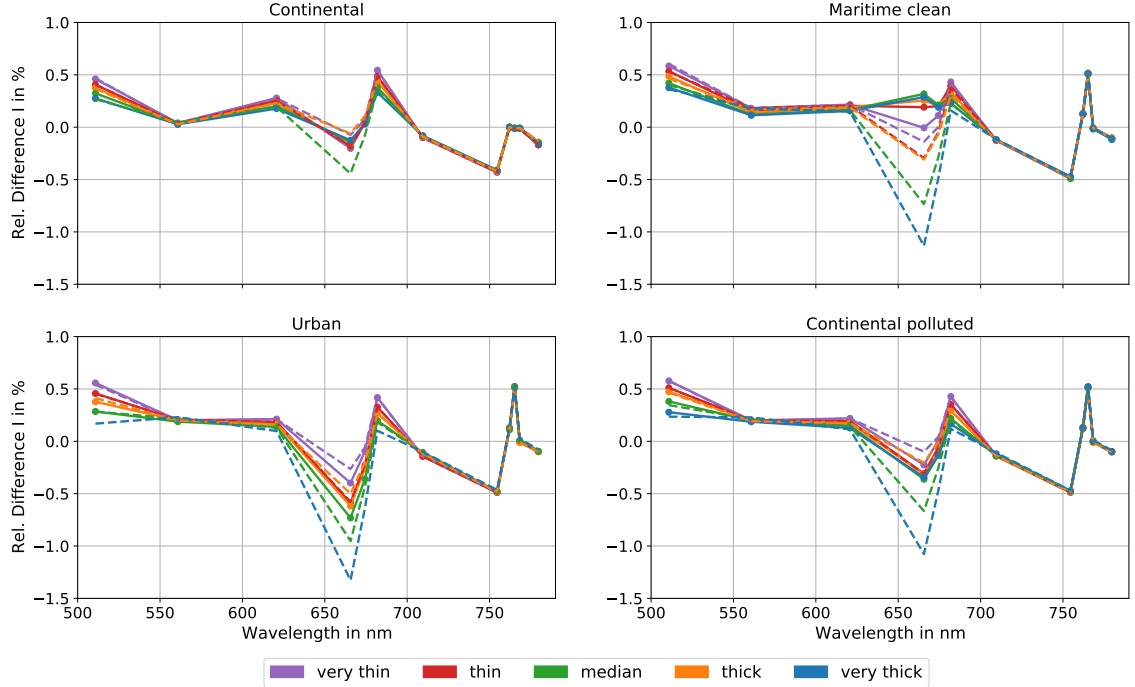

**Figure 10.** Relative difference of radiance after transfer function applied to simulations with different aerosol types. The transfer function was applied using different input parameter. Solid lines: scenario 1, dashed lines: scenario 2. The reference AOD is given in Fig. 8.

tions, the aerosol, surface pressure, gas concentration, measurement geometry, and co-location. Only the *apriori* knowledge of the surface reflectance deviated from the truth. In this case, the OLCI-FLEX surface reflectance is retrieved without error.

In contrast, the interpolated OLCI-A surface reflectance deviates from zero with up to 1.5 % at 510 nm. This deviation shows the limits of the transfer function for the OLCI-FLEX data set. Due to the band distributions of OLCI-FLEX and OLCI-A, a PCR is necessary to allow the interpolation which inserts this residual error. The PCR especially fills the large gaps between 500.625 and 531.875 nm and 620.625 and 681.875 nm.

For the cases of a wrong aerosol characterization, the retrieved surface reflectance at OLCI-FLEX bands deviates strongly

from the truth for all three aerosol models. The strongest deviations are noticed in the case of the maritime clean aerosol whose optical properties deviate strongest from the continental aerosol model (see Fig. 8). Accordingly, the interpolated OLCI-A surface reflectance deviates from the truth. Those effects increase with the AOD. Across the spectrum the absolute difference in the surface reflectance (not shown here) follows the shape of the surface reflectance with an increase at the red-edge. With the larger values of the true surface reflectance for wavelengths of 700 nm and more, this dependency is not shown for the

relative difference due to its scaling.

In scenario 2 the relative difference of the OLCI-AR surface reflectance is also large. However, it shows slightly different features than in scenario 1. With a wrong characterization of the aerosol extinction the effect of the other aerosol optical properties





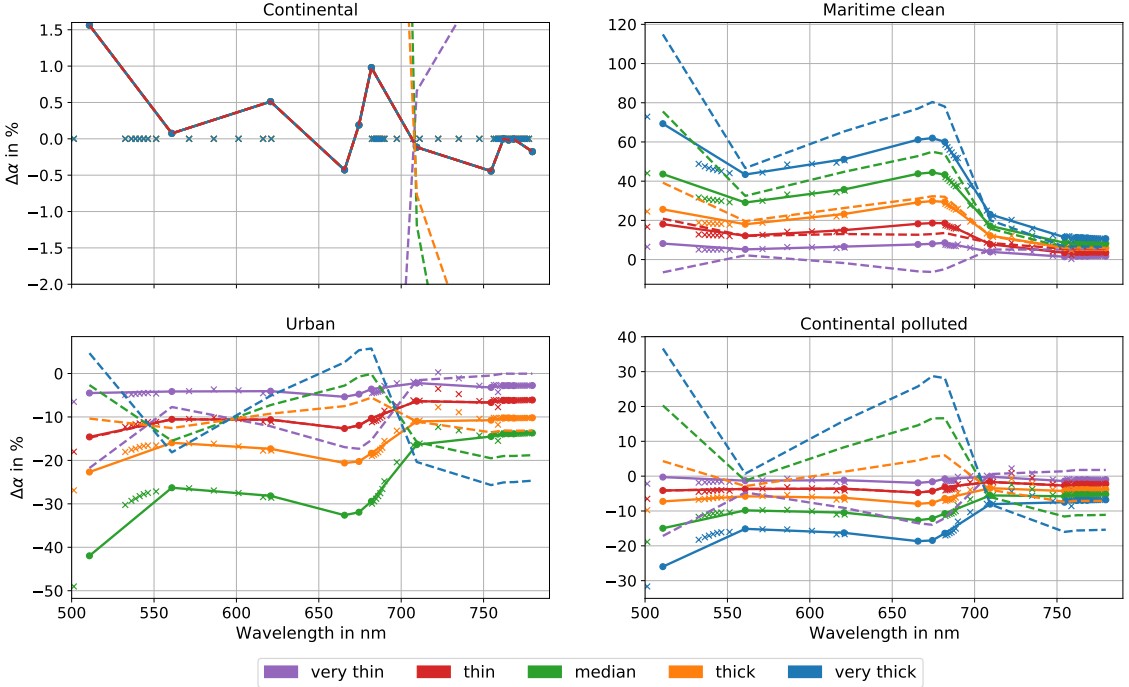

**Figure 11.** Relative difference of optimized/PCR surface reflectance compared to truth for simulations with different aerosol types. The different subplots show optimized surface reflectances with different AOT input. Solid lines: scenario 1, dashed lines: scenario 2. The reference AOD is given in Fig. 8. Crosses mark difference between retrieved surface reflectance at OLCI-FLEX bands and truth.

is either overcompensated ( e.g. in case of the *Continental polluted* model) or increased (e.g. *maritime clean* model).

All in all, the large errors made for the surface reflectance are not translated in the relative difference between OLCI-AR and
OLCI-A TOA radiances, because errors in surface reflectance and aerosol balance each other as long they are used consistently. The goal of the transfer function is not a perfect atmospheric correction and surface retrieval but the estimate of the bias between radiances of two satellites with different spectral responses. This goal is fulfilled as discussed in the previous sections.

### 4.3 Wavelength sensitivity

The central wavelength of OLCI's bands is known with an uncertainty of 0.1-0.2 nm. Figure 12 shows the effect of a 0.1 nm
wavelength shift on the relative difference between reconstructed and measured OLCI-A for data from 2nd July 2018 which were presented in the previous sections. The central wavelength were shifted plus and minus 0.1 nm. Afterwards, the complete transfer function was applied on the measured data, and the relative difference in radiance was compared with the relative difference presented in Fig. 5 by taking the difference of the relative differences. We studied several combinations of wavelength shifts. The most pronounced shifts are presented here. In the first presented case, the OLCI-FLEX central wavelengths are
shifted plus 0.1 nm and the OLCI-A wavelengths minus 0.1 nm. In the second case, we shifted the wavelengths vice versa with



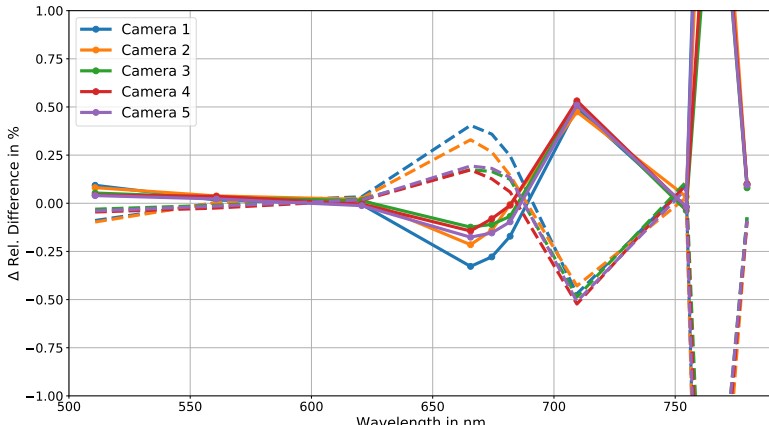

**Figure 12.** Difference of relative difference between reconstructed and measured OLCI-A with the correct central wavelength and shifted central wavelengths. OLCI-FLEX central wavelengths are shifted plus 0.1 nm and OLCI-A is shifted -0.1 nm for the solid lines. Dashed lines show the effect of a wavelength shift of minus 0.1 nm for OLCI-FLEX and plus 0.1 nm for OLCI-A. The y-axis is cut off at +/- 1 % with original maximum values of +/- 5%.

minus 0.1 nm for OLCI-FLEX and plus 0.1 nm for OLCI-A.

The wavelength shifts especially affects the gas absorption bands. Within the oxygen absorption band ($O_2A$ band) around 760 nm the difference is up to 6 %. Even though the OLCI-A bands do not cover the $O_2B$ band between 686 nm and 688 nm, they are indirectly affected by the wavelength shift, because OLCI-FLEX bands cover the $O_2B$ bands. A shift of wavelength

results in a different retrieved surface reflectance and thus a different reconstructed OLCI-A surface reflectance and TOA radiance. The nominal OLCI-A band at 710 nm is affected by the wavelength shift quite strong. It is located in the red edge. A small change in wavelength results in a large difference in the TOA radiance. Shifts with other combinations than the presented cases show similar results.

All in all, the effect of the wavelength shift is mostly visible in absorption bands and is generally small with up to 0.5 %

difference. Only the $O_2A$ band is affected strongly. However, the difference between OLCI-FLEX and OLCI-A of about two percent throughout all cameras and most considered bands does not result from the wavelength uncertainty of 0.1 nm.

## 5   Discussion

### 5.1   Discussion of the results

The application of the transfer function on the OLCI-A/OLCI-FLEX data set of summer 2018 resulted in a direct comparison

of the two data sets. We observed a relative difference in measured TOA radiance between OLCI-FLEX and OLCI-A of about



2 %. OLCI-A measured higher radiances than OLCI-FLEX. A similar difference was observed by Lamquin et al. (2020) when comparing OLCI-A and B with their original band settings. We also found a difference of about 5% with different sign at 778.75 nm for the OLCI-FLEX-OLCI-A comparison which is not observed by Lamquin et al. (2020). Thus, we conclude that it was not caused by an absolute calibration issue between OLCI-A and OLCI-B but by the processing from L0 to L1 of the

OLCI-FLEX data. Furthermore, we observed artefacts for weak absorption bands (708.75 and 767.5 nm) across the field of view of each camera (see Fig. 6). We hypothesize that this is an instrumental effect, maybe it stems from line filling effects due to spectral stray light. This effect was not observed for the OLCI-A/OLCI-B comparison by Lamquin et al. (2020).

The median relative difference is a very robust measure of the overall difference between OLCI-FLEX and OLCI-A. A bootstrap method showed a high representativity of the median. Between 660 and 680 nm, where the PCR was necessary, the

representativity is lower.

The results within $O_2$ absorption bands between 755 and 770 nm have been considered separately. The fine and deep absorption lines of $O_2$ must be described very accurately. The band characterization must be exact with tolerances of less than 0.1 nm. If there are small shifts in the central wavelength or the band width the characterization is not suitable for the band and the according radiance. Furthermore, only with a correct estimate of the surface pressure the depth of the $O_2$ absorption lines can

be simulated correctly. We use the surface pressure from the L1B data which has an uncertainty of $\pm 10$ hPa. In our results, we observe two effects within the $O_2$ absorption bands. In the strong absorbing bands at 761.25 and 764.375 nm of OLCI-A the median difference between OLCI-A and OLCI-AR fluctuates across the detectors, which could be a result of imprecise wavelength characterization. The fluctuations are much smaller than the ones observed by Lamquin et al. (2020), who did not use the time-dependent wavelength characterization. Hence, the used characterization of the oxygen absorption band is

an improvement. The second effect visible in the results is the strong gradient of the relative difference with the wavelength starting at 761.25 nm. This effect cannot be explained by the $O_2$ band, as both strong and weak absorbing bands (764.375 and 767.5 nm) show a similar relative difference which differs from the overall bias between OLCI-A and OLCI-FLEX of about minus 2%. The last OLCI-A band at 778.5 nm is not influenced by $O_2$ absorption but it shows a relative difference of plus 5%. The change in sign of the relative difference with the wavelength at the edge of the spectrum probably originates in the

processing form L0 to L1 of OLCI-FLEX.

The uncertainty and sensitivity analysis showed that the identified measurement and model uncertainties have only a small effect on the result of the transfer function. They cannot explain the relative difference between OLCI-A and OLCI-FLEX radiances. Thus, we could identify actual systematic differences of measurement from the two instruments during the special configuration.

## 5.2 Discussion of the method

The application of the transfer function on the OLCI-FLEX and OLCI-A data from the Sentinel 3 tandem phase showed a sensitivity to a confirmed systematic bias between OLCI-FLEX and OLCI-A. Additionally, we could reveal processing issues. The success of the transfer function relies on accurate radiative transfer simulations, an accurate spectral characterization and the accurate description of the environment.



The parameter with the strongest impact is the surface reflectance. The surface reflectance especially that of vegetated ground has many spectral features which influence the radiance measurements strongly. To transfer the surface reflectance from one band setting to the other, the spectral features covered by both instruments must be measured by the high-resolution instrument. As this was not the case for OLCI-FLEX setting, for OLCI-A bands between 660 and 680 nm additional information were introduced using a PCR. The quality of the PCR determines the quality of the transfer function and introduces uncertainty

to the method.

Besides the surface, the atmospheric conditions influence the radiance measurement and thus the quality of the method. Gas absorption lines are distinct spectral features that affect only bands with central wavelength close to those features. Within the visible spectrum water vapour and oxygen absorption are most prominent. The depth of the absorption lines depends on the total column water vapour and the surface pressure. Both terms must be well characterized to eliminate misinterpretation of

differences that are only caused by errors in the atmospheric characterization.

In contrast, the aerosol description is less important as it is smooth within the visible spectral range. A wrong characterization of the aerosol results in an over or underestimation of the surface which is continuous over the complete spectrum. Due to the consistency of the assumptions about the environment among OLCI-FLEX and OLCI-A data sets, the possible misinterpretation of the surface has only a small effect on the validity of the estimated bias.

The sensitivity study showed a residual error in the relative difference of up to 0.5 % which originate from the interpolation of the surface. This uncertainty will be reduced when the transfer function is applied to the FLEX mission. The interpolation will be more accurate due to the high resolution and high spectral coverage of FLORIS.

The presented method has two limitations: 1) no pixel by pixel comparison and 2) a direct uncertainty measure is only possible for the lower resolution band set. Instead of a pixel wise comparison, we performed a statistical evaluation to mitigate the

effects of imprecise co-location and missing information between 660 and 680 nm. The study area of Europe is characterised by a heterogeneous surface. Slight misalignment of the pixel causes different TOA signal due to the difference in the surface. Those uncertainties can be reduced taking medians over large numbers of pixels. Homogeneous areas such as desserts or oceans could be studied and used for a pixel by pixel comparison. The OLCI-FLEX data set from the tandem phase in 2018 did not cover such areas. Thus, the transfer function could not be applied on measurements over desserts or oceans.

The second limitation is that the uncertainty estimate of the lower spectral resolution instrument cannot be transferred back to the high-resolution spectrum. Thus, only an overall estimate of the agreement of the two instruments is possible. In case of OLCI-A and OLCI-FLEX, OLCI-A is a very well characterised and validated instrument. A bias between OLCI-A and OLCI-B is known. Consequently, we assume that the radiometric calibration of OLCI-FLEX is correct and the observed difference to OLCI-A corresponds to the difference between OLCI-A and OLCI-B. The exception is the bias for wavelengths larger than

760 nm.



# 6 Conclusions

In this article, we showed systematic differences between OLCI-A and OLCI-FLEX during the tandem phase of Sentinel 3A and 3B. The comparison is sensitive to measuring errors and processing issues. In this paper, we showed the application of the transfer function for comparing measurements of satellites flying in tandem formation. An advantage of a tandem mission is the observation of the same geographic target under the same environmental conditions. Further tandem missions are planned for which the transfer function could be applied. One of them is another Sentinel 3 tandem mission. This mission could be used to study the settings of FLEX even further. With reprogrammed band settings that better cover the original OLCI settings the PCR can be omitted and the surface reflectance can be described more robust. An other tandem mission is the coming FLEX-Sentinel-3 mission. The transfer function can be used for the quality control of especially the lower resolution bands between 500 and 677 nm. The higher resolution bands can be most probable directly convolved with OLCI spectral response functions to be comparable with the corresponding OLCI radiance measurements.

The method is not limited to tandem missions. Satellite-satellite comparisons for satellites with different overpass times can be compared too. The requirements for such comparison are a well described atmosphere and surface for both overpasses, an accurate spectral characterization of both instruments and knowledge about the observation and sun angles that need to be considered in the radiative transfer simulations. Additionally, bidirectional surface reflectance effects must be taken into account. When all those requirements are fulfilled, the transfer function allows a comparison among satellites with different spectral settings which can be conducted anytime and which does not need certain target areas. A constant quality check between two instruments with different band settings is possible and thus an inter-operational product can be generated and quality controlled. With such inter-operational products, we can exploit the potential information content of the exciting satellites even further (Niro et al., 2021).

Beside the comparison of between satellite-satellite data, further applications of the transfer function are possible. It can be applied to transfer information gained at TOA from satellites down to BOA data or vice versa. Thus, a comparison of satellite based with ground-based or aeroplane-based radiance measurements is possible. During the Sentinel 3A and B tandem phase in 2018, simultaneous experiments were realized comprising both ground and airborne measurements. The introduced transfer function could be applied on this data set to increase the level of quality control of the described data set.

*Data availability.* The OLCI-FLEX data set can be requested under https://doi.org/10.5270/ESA-624426c.

*Author contributions.* Conceptualization, R.P. and L.J.; methodology, R.P. and L.J.; software, L.J. and R.P.; validation, L.J.; investigation, L.J.; data, M.C., M.T. D.S. and M.D.; writing—original draft preparation, L.J.; writing—review and editing, R.P., M.C., M.T., D.S.; visualization, L.J.; supervision, J.F. and R.P.; project administration, J.F.; funding acquisition, J.F. and R.P. All authors have read and agreed to the published version of the manuscript.



*Competing interests.* The authors declare that they have no conflict of interest.

*Acknowledgements.* The studies presented in this paper were conducted as part of the ESA founded project "Sentinel3 FLEX Tandem Processing Experiment" SF-TAPE. We thank the ESA for the funding. The LUTs for the temporal evolution of the spectral characteristics have been prepared by Rene Preusker as part of the S3MPC under contract to ESA and funded by EC Copernicus budget.



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
