# Peer review of "OLCI A/B tandem phase: Evaluation of FLEX like radiances and estimation of systematic differences between OLCI-A and OLCI-FLEX"

_EGUsphere, 2023_

## Author Response (AR1)

**1 Comments from Referee**

**1.1 Referee 1**

This paper presents a method to compare the OLCI-A and OLCI-FLEX radiometry from the tandem phase acquisitions of Sentinel-3A and Sentinel-3B, paving the way for comparing OLCI and FLEX measurements when FLEX is launched in tandem formation with Sentinel-3. The method is generic and can be extended to other missions, it is well described and the results are well presented.

Compared to the OLCI-A/OLCI-B tandem phase analysis in Lamquin et al., the focus is here made on the spectral range of interest for FLEX between 500 and 800 nm as well as on the vegetated targets of specific interest for FLEX. The commonalities and differences between the results from the two studys are well discussed and the arguments are clear, especially when focusing on the accuracy of spectral characterization and potential failure of the processing of OLCI-FLEX in the NIR. It could be emphasized a better continuity at camera interfaces although a direct comparison should provide a clearer picture.

The method presented in this paper provides an improvement for the precision and uncertainty assessment of the radiometric comparisons when considering vegetated targets. There could be more emphasis on this point since, on a practical aspects, the method is more complex than the one from Lamquin et al.

I recommend this paper to be published after these minor points are considered, as well the points listed below.

Thank you.

line 3 « with high spectral resolution », add « spectral »

line 4 « spatially co-registered measurements…» add « spatially »

line 5 « to describe the atmosphere and the surface» is a bit vague, if you can find something a bit more explicit it's better

line 10 : « The resulting reconstructed low resolution radiance is representative for the high resolution data and it can be compared with the measured low resolution radiance », you could detail it into « The resulting reconstructed low resolution radiance is representative for the high resolution data (OLCI-B measurements in FLEX mode) and it can be compared with the measured low resolution radiance (OLCI-A measurements) ».

line 12 : « for most bands of the OLCI-FLEX spectral domain »

line 12 : « at the longer wavelengths (> 700nm) », or change 700 as desired but provide a boundary

line 18 : « One application could be the quality control of the FLEX mission. » à « One application could be the quality control of the FLEX mission, presently it is also useful for the quality control of the OLCI-FLEX data ». If I understand correctly.

line 23 : « Currently, two twin Sentinel-3... », or add information above that there are multiple instances of S3 satellites which are similar by design but not strictly similar after factory (for instance in spectral characterization)

line 28 : the correct reference for the comparisons of radiances while in tandem is the « part 1 » paper, not the « part 3 » :

Lamquin, N.; Clerc, S.; Bourg, L.; Donlon, C. OLCI A/B Tandem Phase Analysis, Part 1: Level 1 Homogenisation and Harmonisation. Remote Sens. **2020**, 12, 1804. https://doi.org/10.3390/rs12111804

line 37 : « For a meaningful usage of the OLCI-B data in FLEX configuration », precise OLCI-B

line 42 : « It is applied for vegetated cloud free land pixels, as the main objective of FLEX mission is to retrieve fluorescence emitted by plants » : this is a very important methodological point because Lamquin et al. (part 1) did comparisons on different targets and concluded that clouds and deserts are the « easiest » targets for the exercise (the least contaminated by dependency with transmission and other spectral effects). Hence I would add in the following :

line 43 : « Lamquin et al. (2020) showed a systematic bias between OLCI-A and OLCI-B in the tandem constellation data » -> « Lamquin et al. (2020) showed a systematic bias between OLCI-A and OLCI-B in the tandem constellation data, with slight discrepancies depending on the nature of the targets which are likely due to residuals from the spectral alignment of the compared measurands ».

line 44 : « will be estimated by using our transfer function » à « will be estimated by using our transfer function on vegetated pixels» (If I understand correctly, otherwise remove my comment)

line 55 : I suggest rather « similar spatial resolution and observation geometry » instead of « the same spatial resolution and similar observation geometry, since later you speak of « similar spatial resolution » (l 60), also I think « similar » is more appropriate as you may be able to compare slightly different resolutions without having much side effects

line 57 : « to calculate »

line 59 : « instruments flying in tandem » is more general and preferable as tandem could be temporary only and not be a « tandem mission » as OLCI/FLEX

line 69 : « we developed a transfer function. » I would suggest to describe a bit more the purpose as I did not get it at first, following the graph it seems like you could say : « we developed a transfer function allowing to compare gas-corrected TOA radiances on the same spectral setting (resolution and wavelength) », something like that to your convenience...

line 75 : « The reconstructed OLCI-A spectrum will be referred to as OLCI-AR from now on », since the reconstructed spectrum is originally from OLCI-B measurements there can be a confusion with the fact that you align spectrally to OLCI-A as well (or do you align both OLCIs on the same spectral wavelength, what is actually done ?). Maybe calling it « OLCI-B2AR » as in B-to-A-reconstructed could help keeping track of the fact that the original radiometry (and its residual difference against OLCI-A) is from OLCI-B.

Or put some info at line 90-91 like: « The resulting OLCI-AR radiance is representative for the OLCI-FLEX measurement (hence originally from the OLCI-B calibrated radiance)»

And/or : « The difference between the reconstructed and measured OLCI-A radiance quantifies the bias between the two data sets (hence the bias between OLCI-A and OLCI-B radiance, which is why our results can be compared to Lamquin et al.)

After all, I did not get exactly what spectral setting the two datasets are aligned to, the OLCI-A one ? Or a central-wavelength smile-corrected one, identical for both ? Did you try a sensitivity analysis on this (especially for the O2 bands) ?

line 93 : « Besides the L1b radiance of OLCI-A and OLCI-FLEX, …», add the comma at the end

line 98 : « 520 detector rows are aligned along track » : for the spectral dimension I guess as OLCI is push-broom, could you precise ?

line 103 : « The central wavelength are taken from the temporal evolution model of the wavelength characterization » à The central wavelengths (s added)

Table 4 : step could be added on the table for visibility, eventually avoiding to describe it in the text

line 330 : « Only at longer wavelength 330 (780 nm) OLCI-FLEX is darker » à « Only at longer wavelength 330 (780 nm) OLCI-FLEX is brighter »

line 370 : « 370 camera 5 explains the difference of the median over the camera compared to the other cameras seen in Fig. 5. », you could add « , it is interesting to see that a better continuity between cameras 4 and 5 is observed compared to Lamquin et al. ». I'm wondering what could be the cause : updated spectral characterization ? The method itself ? Did you try comparing over cloudy pixels « raw » from your statistics (without using transfer method) ? It could help understanding why.

Note : you could relate this comment to the discussion part

line 371 : « Further camera effects can be observed for all cameras at 708.75 nm. This band is influenced by water vapour absorption » this is not observed in Lamquin et al. specifically in this channel. Could it be due to the spectral resampling of OLCI-FLEX into OLCI « nominal » configuration coupled to $H_2O$ absorption sensitivity ? Is it what you mean ?

line 468 : « affects » instead of « effects » ?

line 481 : « of more than », not « then »

line 515 : « as long as »

line 597-599 : « deserts » not « desserts » J

line 607 : « In this article, we showed systematic differences between OLCI-A and OLCI-FLEX during the tandem phase of Sentinel 3A and 3B » you could add « , consistent with known radiometric differences between OLCI-A and OLCI-B».

line 624 : « of the exciting satellites », Earth Observation science is indeed exciting but I guess you meant « existing » right ? J

**1.2 Referee 2**

The reviewed manuscript is devoted to developing an approach for inter-comparison of the space-borne instruments with different spectral response functions. In particular, using measurements from

OLCI A/B tandem phase, the transfer function was developed for mimicked FLEX high spectral and OLCI A low resolution co-registered measurements.

The manuscript provides the detailed description of the approach for the transfer function. Applying the transfer function on OLCI-A and OLCI-B/FLEX mimicked measurements, estimated systematic difference was found to be similar to the one observed previously for OLCI-A and OLCI-B with their original band settings.

The developed approach can be used for future constellation of FLEX, OLCI A and B instruments where radiometric consistency is crucial for advanced aerosol and surface characterization.

Specific comment.

It is stated that the developed approach for the transfer function "enables direct comparison of instruments with different spectral responses even with different observation geometries …".

It was shown in the paper that the surface reflectance is the main source of uncertainties of the approach. It is also known that land surface can show very strong angular dependence of the reflectance. In this regards, the discussion about the dependence of the transfer function on surface reflectance angular anisotropy (BRDF effect) is necessary.

I recommend the manuscript for publication after minor revision.

**2 Answers from authors to referees**

**2.1 Answer to Referee 1**

Dear Reviewer,

thank you for your helpful comments. We considered most of them in our updated manuscript. We think your comments helped us to improve the manuscript. Below we reply to each of your comments:

Reviewer: It could be emphasized a better continuity at camera interfaces although a direct comparison should provide a clearer picture. We discussed the better continuity among the cameras as you suggested below.

Reviewer:  The method presented in this paper provides an improvement for the precision and uncertainty assessment of the radiometric comparisons when considering vegetated targets. There could be more emphasis on this point since, on a practical aspects, the method is more complex than the one from Lamquin et al. Thank you, we emphasized this point a little bit more in the introduction: "The spectral signature of vegetated surfaces is very complex and thus a method to compensate the differences in spectral response among OLCI-A and OLCI-FLEX is particularly important for those targets." (Line 46-48)

Reviewer comment list:

line 3 « with high spectral resolution », add « spectral » Done.

line 4 « spatially co-registered measurements…» add « spatially » Done.

line 5 « to describe the atmosphere and the surface» is a bit vague, if you can find something a bit more explicit it's better. We give some example parameter to describe it more specific: " for the atmospheric correction and the retrieval of surface parameters e.g. the fluorescence or the leaf area index."

line 10 : « The resulting reconstructed low resolution radiance is representative for the high resolution data and it can be compared with the measured low resolution radiance », you could detail it into « The resulting reconstructed low resolution radiance is representative for the high resolution data (OLCI-B measurements in FLEX mode) and it can be compared with the measured low resolution radiance (OLCI-A measurements) ». Done.

line 12 : « for most bands of the OLCI-FLEX spectral domain » Done.

line 12 : « at the longer wavelengths (> 700nm) », or change 700 as desired but provide a boundary Done.

line 18 : « One application could be the quality control of the FLEX mission. » à « One application could be the quality control of the FLEX mission, presently it is also useful for the quality control of the OLCI-FLEX data ». If I understand correctly. That is correct. We added this sentence.

line 23 : « Currently, two twin Sentinel-3… », or add information above that there are multiple instances of S3 satellites which are similar by design but not strictly similar after factory (for instance in spectral characterization) Done.

line 28 : the correct reference for the comparisons of radiances while in tandem is the « part 1 » paper, not the « part 3 » :

Lamquin, N.; Clerc, S.; Bourg, L.; Donlon, C. OLCI A/B Tandem Phase Analysis, Part 1: Level 1 Homogenisation and Harmonisation. Remote Sens. **2020**, 12, 1804. https://doi.org/10.3390/rs12111804 . Yes, you are absolutely right, thank you.

line 37 : « For a meaningful usage of the OLCI-B data in FLEX configuration », precise OLCI-B. Done.

line 42 : « It is applied for vegetated cloud free land pixels, as the main objective of FLEX mission is to retrieve fluorescence emitted by plants » : this is a very important methodological point because Lamquin et al. (part 1) did comparisons on different targets and concluded that clouds and deserts are the « easiest » targets for the exercise (the least contaminated by dependency with transmission and other spectral effects). Hence I would add in the following :

line 43 : « Lamquin et al. (2020) showed a systematic bias between OLCI-A and OLCI-B in the tandem constellation data » -> « Lamquin et al. (2020) showed a systematic bias between OLCI-A and OLCI-B in the tandem constellation data, with slight discrepancies depending on

the nature of the targets which are likely due to residuals from the spectral alignment of the compared measurands ». Done. We added "Lamquin et al. (2020) showed a systematic bias between OLCI-A and OLCI-B in the tandem constellation data, with slight discrepancies depending on the nature of the targets" but without the interpretation, as we think it does not help for further motivation or understanding.

line 44 : « will be estimated by using our transfer function » à « will be estimated by using our transfer function on vegetated pixels» (If I understand correctly, otherwise remove my comment) Done.

line 55 : I suggest rather « similar spatial resolution and observation geometry » instead of « the same spatial resolution and similar observation geometry, since later you speak of « similar spatial resolution » (l 60), also I think « similar » is more appropriate as you may be able to compare slightly different resolutions without having much side effects Done.

line 57 : « to calculate » Done.

line 59 : « instruments flying in tandem » is more general and preferable as tandem could be temporary only and not be a « tandem mission » as OLCI/FLEX Done.

line 69 : « we developed a transfer function. » I would suggest to describe a bit more the purpose as I did not get it at first, following the graph it seems like you could say : « we developed a transfer function allowing to compare gas-corrected TOA radiances on the same spectral setting (resolution and wavelength) », something like that to your convenience… Done.

line 75 : « The reconstructed OLCI-A spectrum will be referred to as OLCI-AR from now on », since the reconstructed spectrum is originally from OLCI-B measurements there can be a confusion with the fact that you align spectrally to OLCI-A as well (or do you align both OLCIs on the same spectral wavelength, what is actually done ?). Maybe calling it « OLCI-B2AR » as in B-to-A-reconstructed could help keeping track of the fact that the original radiometry (and its residual difference against OLCI-A) is from OLCI-B. We agree that this abbreviation is less confusing. We changed all of them in the manuscript.

Or put some info at line 90-91 like: « The resulting OLCI-AR radiance is representative for the OLCI-FLEX measurement (hence originally from the OLCI-B calibrated radiance)»

And/or : « The difference between the reconstructed and measured OLCI-A radiance quantifies the bias between the two data sets (hence the bias between OLCI-A and OLCI-B radiance, which is why our results can be compared to Lamquin et al.)

After all, I did not get exactly what spectral setting the two datasets are aligned to, the OLCI-A one ? Or a central-wavelength smile-corrected one, identical for both ? Did you try a sensitivity analysis on this (especially for the O2 bands) ? Thank you. We try to make it clearer. Additionally, we added the sentence "To summarize the method, we shift the OLCI-FLEX radiance (measured by OLCI-B) to the band characteristics of OLCI-A using radiative transfer simulations and the OLCI-A spectral response functions." We hope that makes the whole method clearer.

line 93 : « Besides the L1b radiance of OLCI-A and OLCI-FLEX, …», add the comma at the end Done.

line 98 : « 520 detector rows are aligned along track » : for the spectral dimension I guess as OLCI is push-broom, could you precise ? Done.

line 103 : « The central wavelength are taken from the temporal evolution model of the wavelength characterization » à The central wavelengths (s added) Done.

Table 4 : step could be added on the table for visibility, eventually avoiding to describe it in the text That is a good idea. We added the step width in the table.

line 330 : « Only at longer wavelength 330 (780 nm) OLCI-FLEX is darker » à « Only at longer wavelength 330 (780 nm) OLCI-FLEX is brighter » Thank you, done.

line 370 : « 370 camera 5 explains the difference of the median over the camera compared to the other cameras seen in Fig. 5. », you could add « , it is interesting to see that a better continuity between cameras 4 and 5 is observed compared to Lamquin et al. ». I'm wondering what could be the cause : updated spectral characterization ? The method itself ? Did you try comparing over cloudy pixels « raw » from your statistics (without using transfer method) ? It could help understanding why. We added in Section 3.2.2 "However, across the cameras we observe a good continuity with only small discrepancies between the cameras." And in the discussion: "We found the same or a better continuity of the bias across the cameras, especially between Camera 4 and 5, compared to Lamquin et al. (2020). This improvement could be the result of the improved spectral characterization of OLCI-A and OLCI-B using the time evolved spectral model."

Note : you could relate this comment to the discussion part

line 371 : « Further camera effects can be observed for all cameras at 708.75 nm. This band is influenced by water vapour absorption » this is not observed in Lamquin et al. specifically in this channel. Could it be due to the spectral resampling of OLCI-FLEX into OLCI « nominal » configuration coupled to $H_2O$ absorption sensitivity ? Is it what you mean ? We don't know the reasons for those artefacts for sure. I changed the discussion slightly to stress the need for further investigations: "Those artefacts can have different reasons which neither of them is proven. Amongst other the artefacts might result from instrumental effects like a line filling due to straylight, a wrong characterization of the absorbing gas within the method or spectral characterization. Our observations are an interesting finding and should be investigated in further studies."

line 468 : « affects » instead of « effects » ? Done.

line 481 : « of more than », not « then » Done.

line 515 : « as long as » Done.

line 597-599 : « deserts » not « desserts » Done .

line 607 : « In this article, we showed systematic differences between OLCI-A and OLCI-FLEX during the tandem phase of Sentinel 3A and 3B » you could add « , consistent with known radiometric differences between OLCI-A and OLCI-B». Done.

line 624 : « of the exciting satellites », Earth Observation science is indeed exciting but I guess you meant « existing » right ? Done :)

**2.2 Answer to Referee 2**

Dear Reviewer,

thank you for your comment on our manuscript. You asked us to specify the need for a angular dependent description of the surface in case of comparing two measurements with different viewing geometries. We agree, that in this case the surface reflectance should be described using the BRDF. We mentioned this in the conclusion of the manuscript (line 620 in original/ line 638 in updated manuscript).

Best regards,

Lena Jänicke

**3 Changes in the manuscript**

All changes in the manuscript are listed line-by-line in the response to referee 1 printed in green (see 2.1).